# Learning Video-Conditioned Policy on Unlabelled Data with Joint Embedding Predictive Transformer

**Hao Luo**[1] **and Zongqing Lu**[1,2]*
[1]School of Computer Science, Peking University
[2]Beijing Academy of Artificial Intelligence

## Abstract

The video-conditioned policy takes prompt videos of the desired tasks as a condition and is regarded for its prospective generalizability. Despite its promise, training a video-conditioned policy is non-trivial due to the need for abundant demonstrations. In some tasks, the expert rollouts are merely available as videos, and costly and time-consuming efforts are required to annotate action labels. To address this, we explore training video-conditioned policy on a mixture of demonstrations and unlabeled expert videos to reduce reliance on extensive manual annotation. We introduce the Joint Embedding Predictive Transformer (JEPT) to learn a video-conditioned policy through sequence modeling. JEPT is designed to jointly learn visual transition prediction and inverse dynamics. The visual transition is captured from both demonstrations and expert videos, on the basis of which the inverse dynamics learned from demonstrations is generalizable to the tasks without action labels. Experiments on a series of simulated visual control tasks evaluate that JEPT can effectively leverage the mixture dataset to learn a generalizable policy. JEPT outperforms baselines in the tasks without action-labeled data and unseen tasks. We also experimentally reveal the potential of JEPT as a simple visual priors injection approach to enhance the video-conditioned policy.

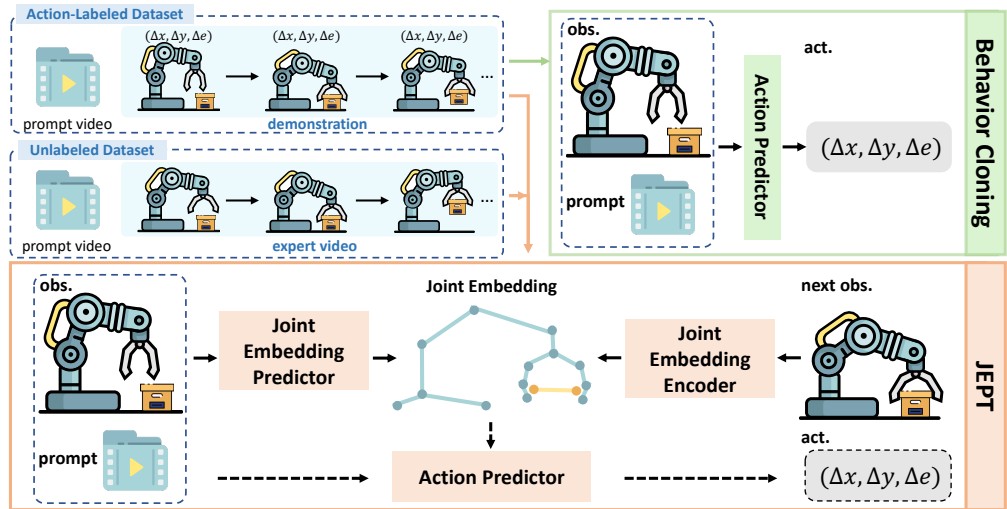

Figure 1: Illustration of the BC and JEPT for visual-conditioned policy learning. Video-conditioned Behavior Cloning (BC) (right top) directly models the policy from the demonstrations. JEPT (bottom) decomposes imitation learning into visual transition prediction (solid arrows) and inverse dynamics learning (dashed arrows). BC is constrained to the action-labeled dataset, whereas JEPT enables visual transition prediction on unlabeled data and leverages the mixture dataset.

*Correspondence to Zongqing Lu <zongqing.lu@pku.edu.cn>.

## 1 INTRODUCTION

Learning generalizable policies through multi-task imitation learning remains a considerable challenge. A generalizable policy should adapt seamlessly to novel tasks whose demonstrations are absent from the training dataset. In response to this, the video-conditioned policy (Jiang et al., 2023; Bahl et al., 2023; Shah et al., 2023) has garnered attention for its superior potential to generalize across tasks. A video-conditioned policy takes a prompt video depicting the task as a condition and executes the desired task in the dynamics encountered. Utilizing a prompt video as a policy condition exhibits a better potential for generalization compared to other forms of task specifications, such as instructions or goals given in language (Brohan et al., 2023; Padalkar & et al., 2023; Jiang et al., 2019; Yenamandra et al., 2023) or image (Bousmalis et al., 2024; Lee et al., 2020; Du et al., 2024; Nair et al., 2018) formats. The abundant information conveyed in videos offers sufficient policy guidance for novel tasks, thus circumventing the misleading caused by language ambiguities or the absence of process depiction in static images (Jain et al., 2024).

Although comprehensive information in prompt videos imbues video-conditioned policy with flexibility and generalization, learning such a policy is more than trivial. Imitating the tasks depicted in prompt videos requires models capable of both temporal reasoning to understand the task and fine-grained control to replicate it. To this end, previous methods (Shah et al., 2023; Jain et al., 2024) typically conduct behavior cloning on large datasets of paired prompt videos and expert demonstrations. However, procuring expert demonstrations can be prohibitively costly and time-consuming. In the tasks where expert policies are hard to obtain, we can acquire the videos of the expert rollouts through human intervention, but these videos are devoid of action labels. Conducting behavior cloning in this scenario requires additional efforts to annotate the action labels. This raises the question: Can these expert videos[1] be effectively harnessed without additional annotation, thus alleviating the data collection burden? Accordingly, we explore the possibility of training a generalizable video-conditioned policy on a dataset containing both expert demonstrations and unlabeled expert videos paired with prompt videos.

Unlike methods that directly imitate expert demonstrations, behavior cloning is infeasible on the mixture dataset due to the absence of the action labels from the data samples with expert videos. Intuitively, we decompose the behavior cloning into two synergistic subtasks to fully exploit the mixture dataset of expert demonstrations and expert videos. Specifically, the process of video-conditioned imitation can be broken down into *visual transition prediction* and *inverse dynamics learning*. In the visual transition prediction, the model learns how the prompt videos should manifest in the dynamics of the tasks. In inverse dynamics learning, the model infers the actions required to realize the visual transition. By combining them, the model can learn to predict plausible future observations and then convert them into actions, enabling video-conditioned imitation.

This design potentially suits the setting of mixture dataset. On the one hand, visual transition prediction is more task-specific in visual imitation learning task and can be more sufficiently learned from both expert videos and demonstrations. On the other hand, although the inverse dynamics is merely embedded in the demonstrations, it remains universally applicable across tasks. Consequently, the expert videos could enhance the learning of visual transitions, while the inverse dynamics derived from the demonstrations could generalize to guide the execution of the tasks without action labels.

In light of these insights, we propose the Joint Embedding Predictive Transformer (**JEPT**) to encapsulate both visual transition and inverse dynamics for video-conditioned policy learning. Building on the framework of Decision Transformers (Chen et al., 2021; Lee et al., 2022; Furuta et al., 2022), we employ a video-conditioned Transformer-based architecture to perform sequence modeling, serving as the policy. As shown in Figure 1, we modify the typical behavior cloning sequence into a two-step process. First, JEPT predicts the embeddings of the next observations, thus capturing visual transition. Then, conditioned on these predicted embeddings, JEPT predicts the corresponding actions, thereby learning inverse dynamics. Additionally, we integrate the Joint Embedding Predictive Architecture (JEPA) (LeCun, 2022) into the video-conditioned Transformer. Prior works (Assran et al., 2023; Bardes et al., 2024) have demonstrated that JEPA is an effective visual representation learning approach, compressing visual inputs into predictive features. By incorporating JEPA, we aim to learn

---

[1]For simplicity and distinction, we use 'prompt videos' referring to videos depicting the desired task and used as the policy condition, while 'expert videos' for the visual observation sequence of the expert rollouts.

predictive visual representations of future observations, thereby improving the transfer of inverse dynamics between action-labeled and unlabeled data.

To evaluate the effectiveness of JEPT, we conduct experiments on Meta-World (Yu et al., 2020a) and Robosuite (Zhu et al., 2020). In our Meta-World experiments, JEPT outperforms all the baselines regarding the average success rates of the tasks within the dataset and the unseen tasks. In the Robosuite experiments, we further validate the effectiveness of JEPT in handling tasks where there is a larger discrepancy between prompt videos and task dynamics. Additionally, we explore the potential of JEPT in injecting visual priors via adjusting the inputs of the joint embedding encoder. Our experiments on Meta-World reveal that the optical flow priors can effectively improve the performance. These results demonstrate the effectiveness of JEPT in jointly leveraging the mixture dataset of expert demonstrations and expert videos to train a generalizable video-conditioned policy.

In summary, our contributions are as follows:

- We propose a novel paradigm, JEPT, to learn video-conditioned policies using the dataset containing a mixture of expert videos and demonstrations.
- We design JEPT as a joint embedding predictive architecture to learn an abstract visual representation for learning a generalizable video-conditioned policy.
- We explore the JEPT as an approach of visual priors injection and experimentally find that JEPT can effectively leverage visual prior knowledge and achieve better performance.

## 2 RELATED WORKS

**Video Prompt Policy Learning.** As a densely informative data form, video inherently contains abundant information for task completion when used as a task description. The detailed guidance of the task completion process in the video makes it possible to learn the policy generalizable to novel tasks beyond the training dataset. Some works (Finn et al., 2017; Duan et al., 2017; Yu et al., 2018) employ meta-learning methods to adapt the policy to novel tasks depicted by prompt video, which requires the similarity between the tasks. More recent works focus on learning policy via behavior cloning on datasets of paired prompt videos and expert demonstrations. These works take the prompt videos as a condition input to the policy network and generalize to unseen tasks with corresponding prompt videos. Various auxiliary mechanisms have been explored to achieve generalizable imitation, such as inverse dynamics prediction (Dasari & Gupta, 2021), cross-painting (Chen et al., 2024), skill decomposition (Shin et al., 2024; 2023), hierarchical policy learning (Jain et al., 2023), text-aligned representation (Jang et al., 2022), contrastive video encoder (Chane-Sane et al., 2023) and observation-attentive representation (Jain et al., 2024). Some works (Sivakumar et al., 2022) design object-centric decomposition for some specific manipulation tasks and achieve generalization within the same task category. The core of these methods remains direct behavior cloning, necessitating access to action-labeled demonstration data. Recent works (Jain et al., 2024; Shah et al., 2023; Jiang et al., 2023) construct large-scale datasets for video-conditioned policy learning, evaluating that large-scale and well-aligned data can effectively improve the video-conditioned policy learning in terms of generalization and success rates. However, these works require extensive demonstration annotation efforts, which is costly and time-consuming. Our work basically follows the video-conditioned policy learning setting. Still, we explore how the video-conditioned policy can learn effectively when the dataset is a mixture of expert videos and demonstrations paired with prompt videos.

**Learning from Videos.** Given the relative accessibility of video data compared to action-labeled demonstrations, some works explore leveraging unlabeled videos to aid in policy learning. Despite the absence of action labels, videos contain substantial decision-related knowledge due to their rich temporal information. A path for learning from videos involves self-supervised representation learning on videos, including masked autoencoder (Radosavovic et al., 2023; Xiao et al., 2022; Yang et al., 2024), temporal contrastive learning (Li et al., 2024; Nair et al., 2023), and video prediction (Seo et al., 2022; Luo et al., 2024). These methods aim to learn a compact and informative representation from videos that aid in policy learning. Other approaches focus on learning reward functions (Escontrela et al., 2023; Yu et al., 2020b; Bruce et al., 2023) or value functions (Bobrin et al., 2024; Chang et al., 2022) to expedite online policy learning. Due to the absence of action labels, these methods require fine-tuning on the downstream tasks for adaptation. Additionally, some research utilizes datasets combining videos and demonstrations, learning an inverse dynamics model (IDM) (Baker et al., 2022; Schmeckpeper et al., 2021; Zheng et al., 2023; Zhang et al., 2022; Kim

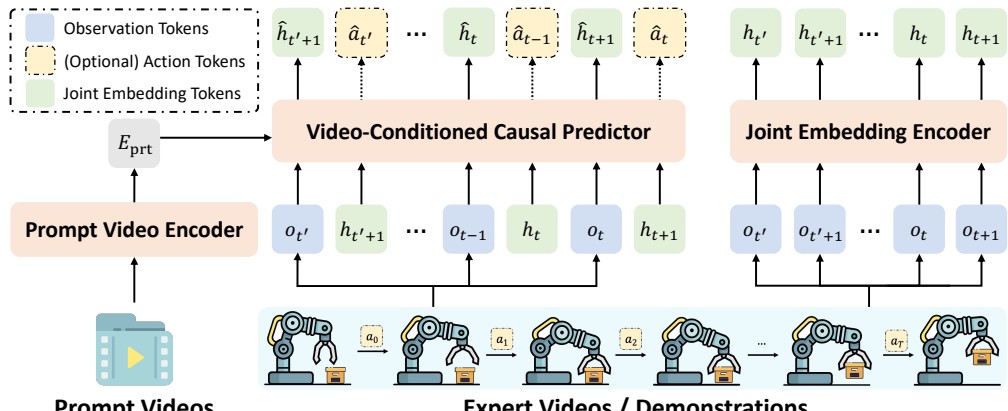

Figure 2: The architecture of the Joint Embedding Predictive Transformer (JEPT). The model consists of a joint embedding encoder, a prompt video encoder, and a video-conditioned causal predictor. The demonstrations and the expert videos are processed as a sequence of tokens, and the causal predictor predicts the tokens causally. The visual embedding token of the next observation is predicted to learn a predictive joint embedding. We use $t' \coloneqq t - k + 1$ to denote the start of the context.

et al., 2023) or a latent inverse dynamics model (Schmidt & Jiang, 2024; Ye et al., 2022; Edwards et al., 2019) on the action-labeled data. The learned IDM can be used to annotate the video data, facilitating imitation learning to derive policies. In these works, the decoupling of inverse dynamics learning and visual imitation may limit the generalizability of the IDM. In our work, however, the learning of visual transitions and inverse dynamics is conducted jointly, offering better generalization.

**Joint Embedding Predictive Architecture.** The Joint Embedding Predictive Architecture (JEPA) (LeCun, 2022) emerges as a promising representation learning framework. JEPA optimizes a predictive loss on the encoded embeddings to learn an embedding space where the embeddings are predictable to each other. Unlike the generative architectures, JEPA optimizes the predictive loss on embeddings rather than raw data, potentially discarding extraneous information to learn more compact and meaningful representations. In order to train a JEPA, it is a common practice to construct data pairs to predict and be predicted. JEPA is a versatile framework. Various works employ different data pair construction methods to enable JEPA to learn embedding spaces tailored to specific information, enhancing tasks such as image classification (Assran et al., 2023), mask classification (Kim et al., 2024), video understanding (Bardes et al., 2024), and motion and content learning (Bardes et al., 2023). Our work integrates JEPA with sequence modeling on trajectories, providing a novel data pair construction method for JEPA. This extends JEPA to architectures akin to the Decision Transformers, aiming to learn visual representations that facilitate inverse dynamics knowledge transfer.

## 3 METHODOLOGY

In this section, we describe how our Joint Embedding Predictive Transformer (JEPT) works on the mixture of expert demonstrations and expert videos paired with prompt videos.

### 3.1 PRELIMINARIES

We aim to learn a video-conditioned policy across a set of tasks. The environment is viewed as a collection of Partially Observable Markov Decision Process variants (POMDPs), with each task represented by a POMDP variant $\mathcal{M}_i \coloneqq (\mathcal{S}, \mathcal{A}, \mathcal{P}, \mathcal{O}, \Omega, \mathcal{V}_i)$. Here, $\mathcal{S}, \mathcal{A}, \mathcal{P}, \mathcal{O}, \Omega$ denote the state space, action space, transition function, observation space, and observation function, respectively, shared across all tasks. Specifically, $\mathcal{V}_i$ comprises a set of prompt videos depicting the task $\mathcal{M}_i$.

**Video-Conditioned Policy.** During interaction with the environment, the agent receives a prompt video $V \in \mathcal{V}_i$ illustrating the desired task at the start of each rollout. At each timestep $t$, the agent obtains a visual observation $o_t \in \mathcal{O}$, derived from the current state $s_t \in \mathcal{S}$ via the observation function $\Omega \colon \mathcal{S} \to \mathcal{O}$. Upon executing an action $a_t \in \mathcal{A}$, the environment transitions to the subsequent

state $s_{t+1}$ according to the transition function $\mathcal{P} \colon \mathcal{S} \times \mathcal{A} \times \mathcal{S} \to [0,1]$. A video-conditioned policy generates actions based on the visual observation context and the prompt video. Formally, a video-conditioned policy is denoted as $\pi\left(a_t | o_{[t-k+1:t]}, V\right)$, where $o_{[t-k+1:t]}$ represents the visual observation context over the last $k$ timesteps.

**Mixture Dataset.** We consider training the video-conditioned policy on a mixture dataset. Ideally, we collect three types of data for imitation learning: (1) the prompt video $V \coloneqq (v_1, v_2, \ldots, v_N)$ specifying the task with $N$ frames, (2) the expert video $O \coloneqq (o_0, o_1, \ldots, o_T)$, the visual observation sequence recorded alongside a $T$-step expert rollout of the task, and (3) the corresponding action sequence $A \coloneqq (a_0, a_1, \ldots, a_{T-1})$ executed during the same rollout. In the tasks where expert demonstrations are available, we assume access to the dataset $\mathcal{D}_{\text{demo}}$, and each data sample $d_{\text{demo}} \in \mathcal{D}_{\text{demo}}$ is in the form:

$$d_{\text{demo}}^{(i)} = \left((v_1, v_2, \ldots, v_N), (o_0, a_0, o_1, a_1, \ldots, o_T) \sim \pi_i^\star\right). \tag{1}$$

Here, $\pi^\star$ represents the expert policies, $\sim$ denotes the rollout process, and $i$ indexes the dataset sample. Conversely, in the tasks where merely videos of expert rollouts are available, we denote this dataset as $\mathcal{D}_{\text{vid}}$, and each data sample $d_{\text{vid}} \in \mathcal{D}_{\text{vid}}$ is in the form:

$$d_{\text{vid}}^{(i)} = \left((v_1, v_2, \ldots, v_N), (o_0, o_1, \ldots, o_T) \sim \pi_i^\star\right). \tag{2}$$

Each data sample in $\mathcal{D}_{\text{demo}}$ consists of a prompt video $V$ and a paired expert demonstration $(O, A)$, while each data sample in $\mathcal{D}_{\text{vid}}$ consists of a prompt video $V$ and a paired expert video $O$. In summary, our objective is to train the video-conditioned policy $\pi\left(a_t | o_{[t-k+1:t]}, V\right)$ on a dataset mixed of these two kinds of data, denoted as $\mathcal{D}_{\text{demo}} \cup \mathcal{D}_{\text{vid}}$.

## 3.2 MODEL ARCHITECTURE

Our JEPT employs a Transformer-based architecture to perform sequence modeling on the data, following the supervised learning paradigm of the Generalized Decision Transformers (Furuta et al., 2022). In this context, demonstrations are treated as sequences of observation and action tokens, where the sequence modeling serves as a video-conditioned policy via predicting the corresponding action tokens. To perform the sequence modeling, JEPT comprises two kinds of modules: (1) visual encoders that process the high-dimensional visual input and (2) a causal predictor aggregating the visual representations to causally predict tokens in the sequence. The comprehensive architecture of JEPT is illustrated in Figure 2, and we elucidate each component in detail below.

### 3.2.1 VISUAL ENCODERS

**Joint Embedding Encoder.** A Joint Embedding Encoder is employed to learn the visual representation for each observation. In our design, the Joint Embedding Encoder captures information from both single-timestep and contextual levels. Structurally, the Joint Embedding Encoder comprises a spatial encoder for single-timestep encoding and a bi-directional temporal encoder for contextual encoding. Formally, the Joint Embedding Encoder $\Psi_{\text{obs}}$ takes as input a visual observation context and outputs the visual representations:

$$(h_{t-k+1}, \ldots, h_{t-1}, h_t) = \Psi_{\text{obs}}(o_{t-k+1}, \ldots, o_{t-1}, o_t). \tag{3}$$

Each representation $h_t$ consists of $N_{\text{obs}}$ tokens. The visual embeddings of the next observation is predicted by the causal predictor and thus a joint embedding will be learned. Via learning joint embeddings predictive of the previous observation tokens, the encoder can compress the high-dimensional visual space into a low-dimensional space and discard extraneous information for effective and generalizable policy learning.

**Prompt Video Encoder.** A Prompt Video Encoder encodes a prompt video into an embedding that serves as a reference for the policy. To specify the desired task, the embedding of the prompt video is used for an implicit understanding of task execution. Specifically, the Prompt Video Encoder comprises a per-frame encoder and a Perceiver Resampler (Jaegle et al., 2022). The Perceiver Resampler aggregates frame embeddings from the per-frame encoder to produce an overall embedding $E_{\text{prt}}$ of the prompt video. Each representation $E_{\text{prt}}$ consists of $N_{\text{prt}}$ tokens. $E_{\text{prt}}$ abstracts task-relevant attributes from videos and serves as the policy condition. Formally, the Prompt Video Encoder outputs as:

$$E_{\text{prt}} = \Psi_{\text{prt}}(v_1, v_2, \ldots, v_N). \tag{4}$$

### 3.2.2 CAUSAL PREDICTOR

Connecting to the visual encoders, a video-conditioned Causal Predictor integrates the visual representations of prompt videos and observations to conduct sequence modeling on the mixture dataset.

To effectively leverage the mixture dataset, the predictor is designed to conduct two subtasks of behavior cloning, visual transition prediction and inverse dynamics learning. The visual transition delineates how observations should evolve to accomplish tasks depicted by the prompt videos. Formally, visual transition prediction approximates the distribution $P\left(o_{t+1}|o_{\leq t}, V\right)$, which is embedded in both $\mathcal{D}_{\mathrm{demo}}$ and $\mathcal{D}_{\mathrm{vid}}$. Through visual transition prediction, the model learns to align the visual observation sequence with the prompt videos. Inverse dynamics learning approximates the distribution $P\left(a_t|o_t, o_{t+1}\right)$, reflecting the actions required to realize a given visual transition. Although the action labels are only available in $\mathcal{D}_{\mathrm{demo}}$, $P\left(a_t|o_t, o_{t+1}\right)$ is shared across tasks due to the shared transition function $\mathcal{P}$. Given the challenge of predicting in the raw visual observation space, we replace $o_{t+1}$ with the joint embedding of the next observation $h_{t+1}$ in seek of a compressed embedding space conducive to knowledge transfer. Accordingly, the causal predictor is tasked with capturing $P\left(h_{t+1}|o_{\leq t}, V\right)$ and $P\left(a_t|o_t, h_{t+1}\right)$.

The input and output sequences of the causal predictor are designed to concurrently capture the visual transition $P\left(h_{t+1}|o_{\leq t}, V\right)$ and the inverse dynamics $P\left(a_t|o_t, h_{t+1}\right)$. As shown in Figure 2, with the video representation $E_{\mathrm{prt}}$ as the prompt, the causal predictor takes the trajectory sequence $(o_{t-k+1}, h_{t-k+2}, \ldots, o_t, h_{t+1})$ as input to causally predict $(\hat{h}_{t-k+2}, \hat{a}_{t-k+1}, \ldots, \hat{h}_{t+1}, \hat{a}_t)$ respectively. Structurally, the predictor comprises a causal Transformer encoder $\Psi_{\mathrm{pred}}$ and two prediction heads, $\Gamma_{\mathrm{obs}}$ and $\Gamma_{\mathrm{act}}$. Respectively, $\Gamma_{\mathrm{obs}}$ predicts the joint embedding of the next observation $h_{t+1}$ from the hidden states of observation tokens $o_t$, and $\Gamma_{\mathrm{act}}$ predicts the action tokens $a_t$ from the hidden states of $h_{t+1}$. Formally, the predictor encodes the input sequence:

$$\boldsymbol{Z} = \Psi_{\mathrm{pred}}\left(E_{\mathrm{prt}}, o_{t-k+1}, h_{t-k+2}, \ldots, o_t, h_{t+1}\right) \tag{5}$$

and predicts the tokens:

$$\hat{h}_{t+1} = \Gamma_{\mathrm{obs}}\left(Z_t^{(o)}\right), \quad \hat{a}_t = \Gamma_{\mathrm{act}}\left(Z_{t+1}^{(h)}\right). \tag{6}$$

Here, $Z_t^{[\cdot]}$ denotes the hidden states of the corresponding tokens in $\boldsymbol{Z}$, while $\hat{h}_{t+1}$ and $\hat{a}_t$ represent the predicted tokens.

In this predictive form, the causal predictor alternatively models $P\left(h_{t+1}|E_{\mathrm{prt}}, h_{\leq t}, o_{\leq t}\right)$ to capture the visual transition and $P\left(a_t|E_{\mathrm{prt}}, o_{\leq t}, h_{\leq t+1}\right)$ to capture the inverse dynamics. During inference, the predictor iteratively feeds the predicted joint embedding tokens back into the input sequence to predict tokens. By combining the two types of token prediction together, the causal predictor indeed works as a planning-based policy, which predicts the desired next observations first and then converts the plan into actions.

### 3.3 TRAINING PROCEDURE

With the designed sequence modeling form, JEPT can simultaneously capture visual transition and inverse dynamics. In this subsection, we describe how JEPT is optimized on the mixture dataset $\mathcal{D}_{\mathrm{demo}} \cup \mathcal{D}_{\mathrm{vid}}$. Specifically, two predictive losses, the visual transition loss and the inverse dynamics loss, are optimized to predict the joint embedding tokens and the action tokens, respectively.

**Visual Transition Loss.** For the joint embedding prediction, we compute the average $L_2$ distance between the predicted joint embedding and the encoded joint embedding as the visual transition loss $\mathcal{L}_{\mathrm{obs}}$ to approximate the visual transition $P\left(h_{t+1}|E_{\mathrm{prt}}, h_{\leq t}, o_{\leq t}\right)$. For both $\mathcal{D}_{\mathrm{demo}}$ and $\mathcal{D}_{\mathrm{vid}}$, the joint embedding of the next observations is available. The visual transition loss $\mathcal{L}_{\mathrm{obs}}$ is defined as:

$$\mathcal{L}_{\mathrm{obs}} = \mathbb{E}_{(V,O)\sim\mathcal{D}_{\mathrm{demo}}\cup\mathcal{D}_{\mathrm{vid}}}\left[\frac{1}{k}\sum_{i=t-k+1}^{t}\left\|h_{i+1} - \hat{h}_{i+1}\right\|_2\right]. \tag{7}$$

**Inverse Dynamics Loss.** For the action prediction, we compute the average Cross-Entropy between the predicted actions and the discrete ground-truth actions as the inverse dynamics loss $\mathcal{L}_{\mathrm{act}}$ to

approximate the inverse dynamics $P\left(a_t | E_{\text{prt}}, o_{\leq t}, h_{\leq t+1}\right)$. The ground-truth action label is merely available in $\mathcal{D}_{\text{demo}}$. The inverse dynamics loss $\mathcal{L}_{\text{act}}$ is defined as:

$$\mathcal{L}_{\text{act}} = \mathbb{E}_{(V,O,A) \sim \mathcal{D}_{\text{demo}}} \left[ \frac{1}{k} \sum_{i=t-k+1}^{t} a_i \log \hat{a}_i \right]. \tag{8}$$

Via optimizing $\mathcal{L}_{\text{obs}}$, JEPT learns to generate the visual transitions aligned with the prompt videos. By optimizing $\mathcal{L}_{\text{act}}$, JEPT learns to convert the visual transitions into actions. Formally, all the components of JEPT are trained with the loss $\mathcal{L}_{\text{total}}$:

$$\mathcal{L}_{\text{total}} = \mathcal{L}_{\text{obs}} + \mathbb{1}_{d \in \mathcal{D}_{\text{demo}}} \cdot c \cdot \mathcal{L}_{\text{act}}, \tag{9}$$

where $c$ is a hyperparameter balancing the two losses, and $\mathbb{1}_{d \in \mathcal{D}_{\text{demo}}}$ is an indicator function that equals 1 when the data sample $d$ is from $\mathcal{D}_{\text{demo}}$ and 0 otherwise. Considering that JEPT predicts the encoded joint embeddings instead of the raw visual observations, we adopt an alternative training procedure to separately optimize the Joint Embedding Encoder and other components in case of potential model collapse. Additionally, an exponential moving average (EMA) of the joint embedding is used when the Joint Embedding Encoder is a fixed target network to stabilize the training, which is widely adapted in the previous JEPAs (Assran et al., 2023; Bardes et al., 2024). The overall process is shown in Algorithm 1 in Appendix A.1.

## 4 EXPERIEMNTS

In this section, we evaluate the effectiveness of our proposed JEPT on the mixture dataset, where the model learns from action-labeled expert demonstrations and unlabeled expert videos. We evaluate JEPT and baselines on two simulated benchmarks, Meta-World (Yu et al., 2020a) and RoboSuite (Zhu et al., 2020). Both benchmarks provide a variety of robotic manipulation tasks and are widely used for evaluating visual control tasks. Via the experiments, we aim to figure out: (1) whether JEPT effectively leverages the additional $\mathcal{D}_{\text{vid}}$ to improve the performance on unlabeled tasks, (2) whether the learned policy can generalize to unseen tasks, and (3) whether the joint embedding predictive mechanism in JEPT is essential for knowledge transfer.

### 4.1 EXPERIMENT SETUP

We first introduce the experimental setup, including the environments, dataset, metrics, and baselines used in our experiments. More details are available in Appendix B.

**Environments.** We replace the language task descriptions in the environments with prompt videos recorded in the environments as task specifications. In our Meta-World experiments, a Sawyer robot arm interacts with various objects. The prompt videos are recorded with the same robot arm performing identical manipulations, which means there is no visual gap between the prompt videos and the expert videos. In our Robosuite experiments, a Panda robot arm interacts with objects. The prompt videos are recorded by performing the same manipulation with various robot arms, including Panda, Sawyer, IIWA, and UR5e, introducing a visual gap between the prompt and expert videos.

**Dataset.** We select 18 tasks for the Meta-World task set and 15 tasks for the Robosuite task set. In order to construct the mixture dataset $\mathcal{D}_{\text{demo}} \cup \mathcal{D}_{\text{vid}}$ and evaluate on the unseen tasks, we split each task set into three subsets: (1) $\mathcal{T}_{\text{demo}}$: the tasks with the prompt videos and the paired expert demonstrations, (2) $\mathcal{T}_{\text{vid}}$: the tasks with the prompt videos and the paired expert videos, and (3) $\mathcal{T}_{\text{unseen}}$: the tasks with merely the prompt videos. For each task, we construct an expert policy in the vector state space and collect demonstrations or videos by running this expert policy in the rendered environments. $\mathcal{D}_{\text{demo}}$ is collected from the tasks in $\mathcal{T}_{\text{demo}}$, while $\mathcal{D}_{\text{vid}}$ is collected from the tasks in $\mathcal{T}_{\text{vid}}$. We also collect prompt videos for the tasks from $\mathcal{T}_{\text{unseen}}$, which are not included in the mixture dataset.

**Metrics.** We evaluate the model performance in terms of success rate, which is the percentage of successful episodes within 50 trials with different random seeds. The success rates of the tasks from $\mathcal{T}_{\text{demo}}$, $\mathcal{T}_{\text{vid}}$ and $\mathcal{T}_{\text{unseen}}$ reflect the model's ability of **behavior cloning**, **learning from videos**, and **one-shot imitation learning**, respectively. We also calculate the average success rates of seen and unseen tasks to evaluate the overall performance.

**Baselines.** The baselines chosen for comparison with JEPT in our setting are as follows:

Table 1: Success Rates (%) of JEPT and the baselines calculated from 50 trials for each task in Meta-World. The average success rate of the tasks in $\mathcal{T}_{demo}$ and $\mathcal{T}_{vid}$ are listed in the first two rows, while the individual success rate of the 4 tasks in $\mathcal{T}_{unseen}$ are listed in the following rows. The average success rates of the seen and unseen tasks are listed in the last two row.

| Task | Vid2Robot-D | Vid2Robot-M | BC+IDM | JEPT+MWM | DT$^\star$ | JEPT |
|---|---|---|---|---|---|---|
| $\mathcal{T}_{demo}$ | 51.8 | 46.3 | 49.5 | 44.5 | **61.3** | 51.3 |
| $\mathcal{T}_{vid}$ | 4.7 | 28.7 | 8.3 | 21.3 | 13.0 | **31.7** |
| **Handle Press** | 10.0 | 22.0 | 8.0 | 18.0 | 6.0 | **28.0** |
| **Lever Pull** | 0.0 | 4.0 | 0.0 | 4.0 | 0.0 | **10.0** |
| **Plate Slide Back** | 4.0 | 8.0 | 0.0 | 2.0 | 0.0 | **14.0** |
| **Faucet Open** | 4.0 | 14.0 | 4.0 | 14.0 | 0.0 | **22.0** |
| **Seen Average** | 28.2 | 37.5 | 28.9 | 32.9 | 37.1 | **41.5** |
| **Unseen Average** | 4.5 | 12.0 | 3.0 | 9.5 | 1.5 | **18.5** |

Table 2: Success Rates (%) of the ablations calculated from 50 trials for each task in Meta-World. We train JEPT on a dataset with various task numbers for $\mathcal{D}_{vid}$ collection. The average success rate of the tasks in $\mathcal{T}_{demo}$ is listed in the first row, while the individual success rate of the 4 tasks in $\mathcal{T}_{unseen}$ are listed in the following rows. The average success rate of the unseen tasks is calculated in the last row.

| Task | None | 2 Tasks | 4 Tasks | 6 Tasks (JEPT) |
|---|---|---|---|---|
| $\mathcal{T}_{demo}$ | **59.5** | 49.8 | 47.0 | 51.3 |
| **Handle Press** | 6.0 | 4.0 | 16.0 | **28.0** |
| **Lever Pull** | 0.0 | 0.0 | 4.0 | **10.0** |
| **Plate Slide Back** | 0.0 | 0.0 | 10.0 | **14.0** |
| **Faucet Open** | 0.0 | 2.0 | 20.0 | **22.0** |
| **Unseen Average** | 1.5 | 1.5 | 12.5 | **18.5** |

- **Vid2Robot** (Jain et al., 2024): conducts behavior cloning and auxiliary representation learning on a large dataset where all the data is action-labeled. We train two variants of Vid2Robot: **Vid2Robot-D** refers to Vid2Robot trained merely on $\mathcal{D}_{demo}$, while **Vid2Robot-M** refers to Vid2Robot trained on $\mathcal{D}_{demo} \cup \mathcal{D}_{vid}$ with behavior cloning loss masked from $\mathcal{D}_{vid}$.

- **BC+IDM**: Zheng et al. (2023); Kim et al. (2023) proposed a paradigm leveraging the mixture dataset for visual imitation learning. These methods separately learn an inverse dynamics model for action annotation and conduct behavior cloning on the mixture dataset. We accommodate these methods to video-conditioned imitation learning.

- **DT$^\star$**: We remove the joint embedding predictive mechanism from JEPT, which indeed conducts behavior cloning with a DT variant on $\mathcal{D}_{demo}$.

- **JEPT+MWM** (Seo et al., 2023): We replace the joint embedding in JEPT with the representation learned with MWM as a baseline. MWM provides a practical representation learning method for visual control tasks. Unlike the joint embedding in JEPT, MWM learns a generative Masked Auto-Encoder (MAE).

## 4.2 META-WORLD EXPERIMENTS

In the Meta-World experiments, we divide the 18 tasks into three subsets: $\mathcal{T}_{demo}$, $\mathcal{T}_{vid}$ and $\mathcal{T}_{unseen}$, respectively containing 8, 6 and 4 tasks.

**Comparisons with Baselines.** The task success rates of JEPT and the baselines are presented in Table 1, where JEPT surpasses all baselines in terms of average success rates. Regarding the success rates of tasks from $\mathcal{T}_{demo}$, JEPT performs slightly worse than DT$^\star$, similar to the gap between Vid2Robot-M and Vid2Robot-D. This may be a side-effect of the more tasks integrated into the policy, leading to a decrease in the performance of individual tasks. Notably, JEPT exhibits a significant advantage in tasks from both $\mathcal{T}_{vid}$ and $\mathcal{T}_{unseen}$, indicating that JEPT effectively leverages the additional $\mathcal{D}_{vid}$ to learn a more generalizable policy. Compared to Vid2Robot-M, the sequence modeling design in JEPT, extracting visual transition and inverse dynamics, might suit the mixture dataset more. Although BC+IDM also captures the inverse dynamics, JEPT integrates transition-aware representation learning with inverse dynamics learning on the mixture dataset instead of an IDM learned merely on $\mathcal{D}_{demo}$, which may capture more generalizable inverse dynamics knowledge and accounts for the better performance of JEPT. Additionally, the outperformance of JEPT over DT$^\star$

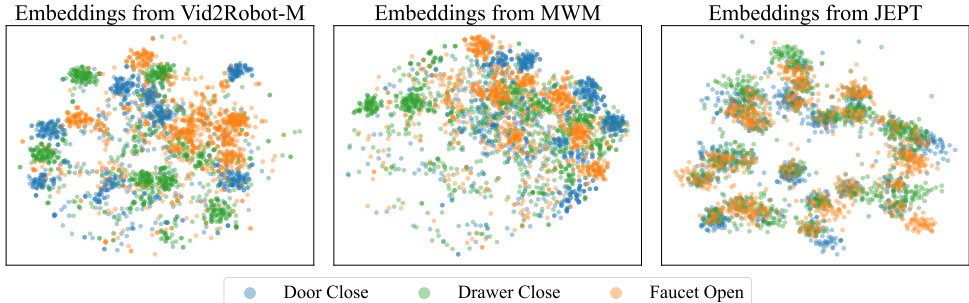

Figure 3: Visualization of the observation embeddings learned in JEPT (right), MWM (mid), and Vid2Robot-M (left). The observation embeddings are calculated in a rollout video from 'Door Close' in $\mathcal{T}_{\text{demo}}$, 'Drawer Close' in $\mathcal{T}_{\text{vid}}$, and 'Facet Open' in $\mathcal{T}_{\text{unseen}}$. We apply t-SNE on one whole embedding set and split the projected vectors according to the algorithms into three sub-figures.

and JEPT+MWM indicates that the predictive joint embedding in JEPT is more suitable for learning on mixture datasets than non-specialized or alternative embedding mechanisms.

**Ablation Study.** We conduct an ablation study to investigate the impact of the size of $\mathcal{D}_{\text{vid}}$. As is listed in Table 2, we train JEPT on datasets where $0, 2, 4$ and $6$ tasks are covered in $\mathcal{D}_{\text{vid}}$. The success rates of tasks increases as the size of $\mathcal{T}_{\text{vid}}$ grows. The results indicates that the additional video data is benificial and JEPT can effectively leverage the data in $\mathcal{D}_{\text{vid}}$ to learn a more generalizable video-conditioned policy. Additionally, when the $\mathcal{D}_{\text{vid}}$ grows from $0$ tasks to $6$ tasks, the performance degradation on the seen tasks might result from a higher integration of the policy, which stands with our observation in Table 1.

**Visualizations.** We also visualize the visual embeddings learned in JEPT, MWM, and Vid2Robot-M. Specifically, we select one task from each of $\mathcal{T}_{\text{demo}}$, $\mathcal{T}_{\text{vid}}$, and $\mathcal{T}_{\text{unseen}}$ and calculate the observation embeddings of a rollout video from each task. We apply t-SNE to the whole set containing the observation embeddings of the three videos from JEPT, MWM, and Vid2Robot to project them into a shared 2D space. As shown in Figure 3, the observation embeddings from Vid2Robot-M and MWM demonstrate severe out-of-distribution phenomena among the action-labeled, unlabeled, and unseen tasks. The embeddings from JEPT exhibit greater similarity in the distribution, potentially contributing to its superior generalization.

### 4.3 ROBOSUITE EXPERIMENTS

We also conduct experiments on the Robosuite, which offers a variety of robotic arms for manipulation tasks. Thus we evaluate the performance of JEPT in the case where there is a visual gap between prompt videos and expert videos. We split the 15 tasks into three subsets: $\mathcal{T}_{\text{demo}}$, $\mathcal{T}_{\text{vid}}$, and $\mathcal{T}_{\text{unseen}}$, containing 6, 5, and 4 tasks, respectively. The results of these experiments are shown in Table 3. Although the success rates in these tasks are somewhat lower than those in the Meta-World experiments due to the increased difficulty, JEPT still outperforms the baselines across all tasks. Similarly to the Meta-world results, the success rate of JEPT in $\mathcal{T}_{\text{demo}}$ is lower than that of DT$^\star$, possibly due to the integration of more tasks into a single policy. However, JEPT still exceeds all the baselines in $\mathcal{T}_{\text{vid}}$ and $\mathcal{T}_{\text{unseen}}$, indicating that JEPT can effectively leverage $\mathcal{D}_{\text{vid}}$ to learn more generalizable policies despite visual gaps.

### 4.4 VISUAL PRIOR INJECTION

Considering that visual priors learned from external datasets may enhance the generalization of the model due to their universality, we additionally explore the possibility of injecting such priors from the pre-trained models into JEPT. A directly injection approach is adopted in our practice. We simply replace the output of the single-timestep spatial encoder in the Joint Embedding Encoder with the output from the visual encoders of some pre-trained visual models. Specifically, we experiment on Meta-World with FlowFormer (Huang et al., 2022), VideoMAE-v2 (Wang et al., 2023), Dino-v2 (Oquab et al., 2024), and SAM (Kirillov et al., 2023), representing visual priors related to optical flow, video reconstruction, depth estimation, and visual segmentation, respectively. As shown

Table 3: Success Rates (%) of JEPT and the baselines calculated from 50 trials for each task in Robosuite. The average success rate of the tasks in $\mathcal{T}_{\text{demo}}$ and $\mathcal{T}_{\text{vid}}$ are listed in the first two rows, while the individual success rate of the 4 tasks in $\mathcal{T}_{\text{unseen}}$ are listed in the following rows. 'X T' refers to the task where Panda robot arm performs the task **T** with the prompt video recorded with **X** robot arm. The average success rates of the seen and unseen tasks are listed in the last two rows.

| Task | Vid2Robot-D | Vid2Robot-M | BC+IDM | JEPT+MWM | DT$^\star$ | JEPT |
|------|------|------|------|------|------|------|
| $\mathcal{T}_{\text{demo}}$ | 26.8 | 20.7 | 26.7 | 24.3 | **36.3** | 27.2 |
| $\mathcal{T}_{\text{vid}}$ | 4.0 | 11.6 | 8.4 | 10.8 | 2.0 | **12.8** |
| **Panda Lift** | 4.0 | 16.0 | 6.0 | 22.0 | 8.0 | **38.0** |
| **Sawyer Lift** | 0.0 | 2.0 | 0.0 | 6.0 | 0.0 | **16.0** |
| **IIWA Lift** | 0.0 | 4.0 | 0.0 | 2.0 | 0.0 | **12.0** |
| **UR5e Lift** | 0.0 | 0.0 | 0.0 | 0.0 | 0.0 | **8.0** |
| **Seen Average** | 15.5 | 16.1 | 17.5 | 17.6 | 19.2 | **20.1** |
| **Unseen Average** | 1.0 | 5.5 | 1.5 | 7.5 | 2.0 | **18.5** |

Table 4: Success Rates (%) of JEPT injected with different visual priors calculated from 50 trials for each task in Meta-World tasks. The average success rate of the tasks in $\mathcal{T}_{\text{demo}}$ and $\mathcal{T}_{\text{vid}}$ are listed in the first two rows, while the individual success rate of the 4 tasks in $\mathcal{T}_{\text{unseen}}$ are listed in the following rows. The average success rate of the seen and unseen tasks are calculated in the last two row.

| Task | JEPT+FlowFormer | JEPT+VideoMAE-v2 | JEPT+Dino-v2 | JEPT+SAM |
|------|------|------|------|------|
| $\mathcal{T}_{\text{demo}}$ | **59.8** | 33.3 | 38.5 | 41.3 |
| $\mathcal{T}_{\text{vid}}$ | **35.3** | 11.7 | 9.3 | 19.0 |
| **Handle Press** | **36.0** | 8.0 | 8.0 | 16.0 |
| **Lever Pull** | **10.0** | 2.0 | 2.0 | 4.0 |
| **Plate Slide Back** | **6.0** | 0.0 | 0.0 | 2.0 |
| **Faucet Open** | **26.0** | 4.0 | 4.0 | 10.0 |
| **Seen Average** | **47.5** | 22.5 | 23.9 | 30.1 |
| **Unseen Average** | **19.5** | 3.5 | 3.5 | 8.0 |

in Table 4, JEPT+FlowFormer outperforms other visual priors and surpasses the original JEPT. This indicates that the optical flow priors are beneficial for visual control tasks and can generalize across tasks. Meanwhile, JEPT can effectively incorporate the optical flow priors to enhance model performance. The poor performance of other visual priors may result from the inherent unsuitability of these priors for visual control tasks or the inadequacy of the injection approach, requiring further exploration in future work.

## 5 CONCLUSION

In this paper, we propose the Joint Embedding Predictive Transformer (JEPT), a novel approach for video-conditioned policy learning. JEPT is designed to learn from a mixture of expert demonstrations and expert videos paired with prompt videos, aiming to reduce the burden of action label annotation. To suit the mixture dataset, we decompose the video-conditioned policy learning into two subtasks: visual transition prediction and inverse dynamics learning. By jointly learning the two subtasks in the sequence modeling, JEPT works as a planning-based policy. We implement JEPT as an extension of the joint embedding predictive architecture to learn an abstract representation of visual observation, which aids in generalizing video-conditioned policy. Experimentally, we evaluate the effectiveness of JEPT on a series of visual control tasks. Additionally, we explore the JEPT as a simple visual priors injection approach and find it valid in injecting optical flow knowledge.

**Limitations.** A series of experimental results of JEPT indicate that joint learning visual transition and inverse dynamics allow it to effectively leverage the mixture dataset and derive a generalizable policy. Despite this, we have also identified some limitations. Since our work focuses on solving problems within datasets containing expert videos, our method does not include additional designs for the potential visual gap between the prompt videos and task dynamics. We have validated the effectiveness of JEPT in addressing a certain level of visual gap in Robosuite environments, but more design and validation are needed to apply JEPT to one-shot visual imitation with more discrepancy prompt videos such as human videos. In this regard, our attempt at injecting visual priors could be one beneficial approach.

ACKNOWLEDGMENTS

This work was supported by NSFC under Grant 62450001 and 62476008. The authors would like to thank the anonymous reviewers for their valuable comments and advice. The authors express special thanks to Xinrun Xu (v.xinrun@gmail.com) for her assistance in the figures of this work.

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

# A  IMPLEMENTATIONS

We build our framework based on PyTorch (Paszke et al., 2019) and use the implementations of Transformer modules from the codebase x-transformers[2]. We use the 3090 Nvidia GPU and i9-12900K CPU for the JEPT training and testing. It takes around 10 hours to train JEPT on the Meta-World dataset and 16.5 hours to train JEPT on the Robosuite dataset. The details of the implementations are as follows.

## A.1  ALGORITHM

---

**Algorithm 1** Joint Embedding Predictive Transformer

---

1: **Input:** Mixture dataset $\mathcal{D}_{\text{demo}} \cup \mathcal{D}_{\text{vid}}$
2: **Initialize:** JEPT components $\Psi_{\text{obs}}, \Psi_{\text{prt}}, \Psi_{\text{pred}}, \Gamma_{\text{obs}}, \Gamma_{\text{act}}$ with random weights
3: **for** $e = 1, 2, \ldots$ **do**
4:     **for** $d \in \mathcal{D}_{\text{demo}} \cup \mathcal{D}_{\text{vid}}$ **do**                    ▷ Optimizing Joint Embedding Encoder
5:         Encode $V$ and $O$ into $E_{\text{prt}}$ and $(h_{t-k+1}, \ldots, h_t)$ as Equations 3 and 4
6:         Predict the joint embedding tokens and action tokens as Equations 5 and 6
7:         Compute $\mathcal{L}_{\text{total}}$ as Equation 9
8:         Update $\Psi_{\text{obs}}$ with $\mathcal{L}_{\text{total}}$
9:     **end for**
10:    **for** $d \in \mathcal{D}_{\text{demo}} \cup \mathcal{D}_{\text{vid}}$ **do**                    ▷ Optimizing other components
11:        Repeat steps in lines 5-7
12:        Update $\Psi_{\text{prt}}, \Psi_{\text{pred}}, \Gamma_{\text{obs}}, \Gamma_{\text{act}}$ with $\mathcal{L}_{\text{total}}$
13:    **end for**
14: **end for**

---

## A.2  MODEL DETAILS

**Joint Embedding Encoder.** To encode visual observations, we utilize a pre-trained ViT (Dosovitskiy et al., 2021) variant, 'vit_base_patch16_224', as the spatial visual encoder. To effectively compress the tokens of visual observations, a Perceiver-IO (Jaegle et al., 2022) with a fixed number of learnable queries is employed to compress the visual tokens of single-frame observations. The ViT weights remain fixed during training, allowing the Perceiver-IO to function as an adaptor network. The compressed visual tokens are concatenated with representations of the self-state vector produced by an MLP to form the observation tokens. A bi-directional Transformer temporal encoder is used to encode the observation tokens with context awareness. For visual prior injection approaches in Section 4.4, we simply replace the outputs of the fixed ViT with the embedding outputs of visual encoders pre-trained to capture visual priors. Notably, there are subtle distinctions in input processing among these four visual prior injections. For FlowFormer, observation tokens at each temporal step are computed from adjacent frames. VideoMAE v2 replicates all video frames excluding the first to preserve the original temporal length. Dino v2 and SAM v2 both utilize single-frame images as inputs. Moreover, these models diverge in architectural design, pre-training datasets, and methodological approaches. Despite these variations, our visual prior experiments do not aim to comparatively evaluate visual prior performance. Instead, our objective is to ascertain whether a joint embedding predictive of these representations with visual priors can further compress visual information and enhance generalization.

**Prompt Video Encoder.** To encode the prompt videos, we utilize the same ViT variant as the Joint Embedding Encoder for encoding the visual tokens. Two cascaded Perceiver-IOs are employed to aggregate these tokens. The first Perceiver compresses the tokens from a single frame, while the second compresses the flattened tokens from the first layer to form the prompt video embeddings. During training, only the Perceiver-IOs are trained, the ViT weights remain fixed.

**Video-Conditioned Causal Predictor.** To process observations in expert videos and demonstrations, we employ the same structure as the Joint Embedding Encoder to encode single-timestep observations. Notably, the single-timestep encoder shares the weights in our implementation, and the EMAs are

---

[2]https://github.com/lucidrains/x-transformers

calculated when producing the target joint embeddings. A causal Transformer is utilized, taking as input the sequence of all compressed tokens. A causal mask, where tokens attend only to preceding tokens and tokens from the same observations, ensures prediction causality. To predict the joint embedding tokens and action tokens, two prediction heads are applied to the causal Transformer's output. Specifically, an MLP predicts the joint embedding tokens, while a cross-attention Transformer with learnable action queries for each action dimension predicts the action tokens.

### A.3 BASELINE ADAPTATION

**Vid2Robot** We implement Vid2Robot following the original paper (Jain et al., 2024), as no codebase is released. Similar to our structure, a spatial visual encoder followed by a Perceiver processes the spatiotemporal information of videos and observation sequences. We use the same ViT variant as the Joint Embedding Encoder to encode the visual tokens of the prompt videos and observations, ensuring consistency in comparison. We align the Perceiver structure and the action prediction head in Vid2Robot with our framework. Since no text instruction is provided in our setting, we remove the auxiliary text-video contrastive loss in Vid2Robot.

**MWM** We adapt the released codebase of MWM (Seo et al., 2023) to our setting. The only difference is the absence of a reward signal in our setting; thus, we remove the reward prediction loss.

**DT$^\star$** We adapt our implementation of JEPT to perform sequence modeling in a typical behavior cloning sequence $P(a_t|E_{\text{prt}}, o_1, a_1, \ldots, o_t)$.

### A.4 HYPERPARAMETERS

In our implementation, the Perceiver-IO structure is shared across the Joint Embedding Encoder, the Prompt Video Encoder, and the Video-Conditioned Causal Predictor. The hyperparameters for the Perceiver-IOs and other modules are detailed in Table 5. During training, $E_{\text{prt}}$ is initially excluded in the causal predictor as a warm-up strategy. Furthermore, the hyperparameters for the training process are listed in Table 6.

## B ENVIRONMENT SETUP DETAILS

### B.1 ENVIRONMENT

**Meta-World Environment.** In the Meta-World environment, a Sawyer robot arm is directed to execute a variety of manipulation tasks. We employ 18 tasks originally from Meta-World to establish our video prompt visual control framework. As shown in Figure 4, for each task, a video of the identical manipulation in the same environment, albeit with a different random seed, is provided as the task prompt, while the original task descriptions are omitted. We utilize the customized *camera1* perspective for visual observation and resize the image from the camera to $224 \times 224$ pixels. In addition to the visual observation, we assume the agent is aware of its current self-state, which encompasses the joint angles, joint velocities, and end-effector position. A vector representing these self-state details is supplied alongside the visual camera image to form the observation. For action prediction using Cross-Entropy, the original continuous action space is discretized into 7 bins for each action dimension. During the rollout of each task, the episode is truncated after 250 steps, with an action repeat of 2.

**Robosuite Environment.** In the Robosuite environment, we select tasks involving a Panda robot arm performing various manipulation tasks. We employ two tasks originally from the Robosuite simulator and four tasks implemented by MimicGen (Mandlekar et al., 2023) within the Robosuite framework to create our video prompt visual task. As illustrated in Figure 4, a video depicting a specific robot arm completing the same manipulation serves as the task prompt. The robot in these prompt videos may be a Panda, Sawyer, IIWA, or UR5e, indicating a potential visual gap between the prompt videos and the task dynamics. Two tasks are considered distinct if the robot arm in the prompt videos or the desired manipulation differs. Consequently, we construct 15 tasks from the 6 tasks of MimicGen. We utilize the 'agentview_image' attribute of the state as the visual image. Similar to the Meta-World environment, the visual image is resized to $224 \times 224$ pixels and accompanied by the self-state vector. The continuous action space is discretized into 7 bins per dimension. During each

Table 5: The hyperparameters of the JEPT Modules.

| Single-Frame Perceiver-IO | |
| --- | --- |
| num latents | 64 |
| num queries | 8 |
| embedding channels | 512 |
| attention heads | 8 |
| encoder cross-attention layers | 1 |
| encoder self-attention layers | 2 |
| decoder cross-attention layers | 1 |
| attention feedforward dim | 1024 |
| attention dropout | 0.1 |
| **Temproal Perceiver-IO** | |
| num latents | 128 |
| num queries | 32 |
| embedding channels | 512 |
| attention heads | 8 |
| encoder cross-attention layers | 1 |
| encoder self-attention layers | 2 |
| decoder cross-attention layers | 1 |
| attention feedforward dim | 892 |
| attention dropout | 0.08 |
| **Bi-directional Tempral Encoder** | |
| joint embedding dim | 256 |
| attention layers | 2 |
| attention heads | 8 |
| attention dim | 512 |
| attention feedforward dim | 1024 |
| attention dropout | 0.1 |
| attention activation | GeLU |
| **Causal Transformer** | |
| Max prompt length | 250 |
| prompt tokens num $N_{prt}$ | 32 |
| obs tokens num $N_{obs}$ | 9 |
| context length $k$ | 8 |
| embedding channels | 584 |
| attention heads | 8 |
| attention layers | 4 |
| attention feedforward dim | 1168 |
| attention dropout | 0.05 |
| Attention Activation | GeLU |
| **Action Prediction Head $\Gamma_{act}$** | |
| Cross-attention layers | 2 |
| Cross-attention heads | 4 |
| Cross-attention dim | 256 |
| Feedforward dim | 512 |
| activation | GeLU |
| **Joint Embedding Prediction Head $\Gamma_{obs}$** | |
| layers | 3 |
| hidden dim | 256 |
| activation | LeakyRelu |

Table 6: The hyperparameters of the training process.

| Training Hyperparameters | Value |
|---|---|
| Minibatch size | 90 |
| Optimizer | AdamW |
| Joint Embedding Encoder Learning Rate | 3e-5 |
| Prompt Video Encoder Learning Rate | 7e-5 |
| Video-Conditioned Causal Predictor Learning Rate | 7e-5 |
| Weight Decay | 1e-4 |
| Max Gradient Clip | 1.0 |
| Warm Up Steps | 200 |
| Loss Weight $c$ | 8 |

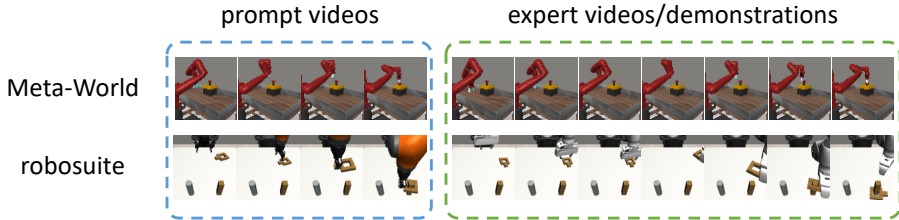

Figure 4: The video prompt task setting of the Meta-World and Robosuite. In Meta-World, the task specification is a video of the same manipulation in the same environment with different random seed. In Robosuite, the task specification is a video of a certain robot arm executing the same manipulation, and the visual gap between the prompt videos and the task dynamics is likely.

task rollout, the episode is truncated after 200 steps, although there is no restriction on episode length within the Robosuite environment.

## B.2 DATASET

**Task Subset.** In both Meta-World and Robosuite, tasks are manually categorized into three subsets: those with expert demonstrations and prompt videos, $\mathcal{T}_{\text{demo}}$; those with expert videos and prompt videos, $\mathcal{T}_{\text{vid}}$; and those with only prompt videos, $\mathcal{T}_{\text{unseen}}$. The specific task divisions are detailed in Table 7 and 8. For Meta-World experiments, 500 expert videos/demonstrations are collected by executing expert policies on tasks from $\mathcal{T}_{\text{demo}} \cup \mathcal{T}_{\text{vid}}$. For Robosuite experiments, 1000 expert videos/demonstrations are gathered for each task in the dataset. For the tasks in $\mathcal{T}_{\text{unseen}}$, 10 prompt videos are sampled for 5 trials with different random seeds on each to assess the one-shot imitation capability of the learned policy. In the ablation study of task numbers covered in $\mathcal{T}_{\text{vid}}$, we use the 'Button Press Topdown Wall' and 'Door Open' tasks for the 2-task variant. Additionally, we include the 'Reach Wall' and 'Drawer Close' tasks for the 4-task variant.

**Expert Policy.** To construct the data used for training and evaluation, we develop expert policies for individual tasks within the vector state space, collecting both prompt videos and corresponding expert demonstrations. Initially, expert policies are formulated in the **continuous** action space. In Meta-World, policies are trained using PPO for each task, utilizing the vector state and reward function provided by the environment. We integrate rule-based policies from the GitHub repository[3] with PPO policies to form expert policies, executing rule-based policies with a probability of 0.85 and PPO policies with a probability of 0.15 to ensure diversity. For tasks in Robosuite, expert policies are learne through behavior cloning on the expert demonstrations from MimicGen. Additionally, expert policies for 'Stack' and 'Lift' tasks are trained using DrQ-v2 (Yarats et al., 2021) with no demonstrations available in Mimicgen. To collect expert rollouts in the **discrete** action space, continuous expert policies are used to generate actions, which are then replaced with the nearest discrete actions in the bins for interaction with the environment. A performance drop is noted in the discrete action space compared to the continuous action space. Specifically, the average success rate for expert policies in

---

[3]https://github.com/Farama-Foundation/Metaworld/tree/master/metaworld/policies

Table 7: The division of tasks in our Meta-World experiments.

| Task Subset | Tasks |
|---|---|
| $\mathcal{T}_{\text{demo}}$ | Button Press, Button Press Topdown, Handle Pull, Reach, Door Close, Window Open, Plate Slide, Faucet Close |
| $\mathcal{T}_{\text{vid}}$ | Button Press Topdown Wall, Door Open, Reach Wall, Drawer Close, Plate Slide Back Side, Window Close |
| $\mathcal{T}_{\text{unseen}}$ | Handle Press, Lever Pull, Plate Slide Back, Faucet Open |

Table 8: The division of tasks in our Robosuite experiments. **'X T'** refers to the task where the Panda robot arm performs the task **T** with the prompt video recorded with **X** robot arm.

| Task Subset | Tasks |
|---|---|
| $\mathcal{T}_{\text{demo}}$ | Panda Stack, Sawyer Stack, IIWA Stack, UR5e Stack, Three Pieces Assembly, Coffee |
| $\mathcal{T}_{\text{vid}}$ | Panda Square, Sawyer Square, IIWA Square, UR5e Square, Stack Three |
| $\mathcal{T}_{\text{unseen}}$ | Panda Lift, Sawyer Lift, IIWA Lift, UR5e Lift |

Meta-World decreases from $91.5\%$ to $85.3\%$ post-discretization, and in Robosuite, it declines from $78.3\%$ to $69.7\%$. For prompt video collection, we record visual observation sequences of the expert rollouts, while for expert videos or demonstrations, both visual observation sequences and self-state vector sequences are recorded.

## C  ADDITIONAL EXPERIMENTS

### C.1  VISUAL PRIOR INJECTION ON ROBOSUITE

In order to supplement the effects of various visual priors on JEPT in different environments, we conduct experiments on visual prior injection within the RoboSuite environment. The implementation of visual prior injection is consistent with the experiments conducted in Meta-World. The results, as shown in Table 9, indicate that the performance of JEPT can be slightly enhanced by utilizing priors based on optical flow to process the visual inputs. This suggests that optical flow might be a suitable prior for enhancing JEPT's performance across different environments. However, all of these visual encoders have been trained on large datasets across numerous visual tasks and possess certain generalizable priors for handling visual inputs. Our experiments with visual priors aim to ascertain whether a joint embedding predictive of these representations, when combined with visual priors, can further compress visual information and improve generalization. From the perspective of introducing more generalizable visual priors, these different priors for processing visual inputs should be logical. However, the success of learning predictive joint embeddings of these priors and effectively enhancing performance may depend on selecting more refined injection methods and better training designs. We merely highlight JEPT's potential in this domain, and further detailed research on other visual prior injections is left for future exploration.

### C.2  ADDITIONAL ABLATIONS

We conduct additional ablation studies to explore the influence of the hyperparameter $c$ in JEPT. The value of $c$ is varied within the set $\{0.3, 1, 5, 8, 12\}$, and the performance of JEPT is assessed in Meta-World. The results are presented in Table 10. When $c$ remains within a reasonable range, JEPT exhibits similar performance. However, when $c$ is excessively small or large, JEPT's performance declines significantly. This underscores the critical role of the hyperparameter $c$ in JEPT's efficacy. In other experiments, we set $c = 8$, as it yields the optimal performance in Meta-World.

To thoroughly investigate the impact of the ratio between $D_{\text{demo}}$ and $D_{\text{vid}}$ on model performance, we conduct additional ablation experiments. Specifically, beyond the original 14-task training dataset,

we incorporate 6 additional tasks from Meta-World to collect training data (Handle Pull Side, Door Lock, Drawer Open, Plate Slide Side, Dial Turn, Handle Press Side). We maintain a constant total amount of training data while varying the ratio of $\mathcal{T}_{\text{demo}}$ to $\mathcal{T}_{\text{vid}}$. For various ratios, we experiment with different hyperparameters $c$, and the results are presented in Table 11. Given the variability of $\mathcal{T}_{\text{demo}}$ and $\mathcal{T}_{\text{vid}}$, we report the average success rate on $\mathcal{T}_{\text{unseen}}$ under different settings. The results indicate that within a certain range, $D_{\text{vid}}$ effectively compensates for the lack of data with action labels. However, when $D_{\text{demo}}$ is insufficient, it may not adequately learn, making IDM difficult to generalize to unseen tasks. Moreover, the results suggest that when $D_{\text{vid}}$ is reduced, a larger $c$ is often required to achieve better performance.

Considering that employing a fixed video encoder mitigates the distribution shift in video representation between action-labeled and unlabeled data, thereby enhancing the generalization of the inverse dynamics model in BC+IDM, we conducted experiments on Meta-World using a pre-trained video encoder, denoted as BC+IDM*. Most pre-trained video encoders utilize a kernel size of 2 for temporal compression, halving the representation length in the temporal dimension. Similar to our approach in visual prior injection with VideoMAE v2, we repeat all frames within the video except for the first frame to maintain consistent length and accurately decode the actions. Employing VideoMAE v2, the experimental results are presented in Table 12. There is a slight improvement in success rates on unlabeled tasks, but no gain on unseen tasks. Overall, the improvement from using a fixed video encoder is limited.

### C.3 ADDITIONAL VISUALIZATION

We conduct a series of visualizations within the Meta-World environment to reveal the distribution of some variants within the rollouts. Specifically, we process 150 videos for each of the 18 different tasks using a prompt video encoder to encode them into $E_{prt}$. We flatten these embeddings and apply t-SNE to reduce the dimensionality to two dimensions. As is shown in Figure 5, there are clear distinctions between different tasks, suggesting that JEPT effectively captures and represents the unique features of each task. Additionally, we observe a degree of similarity between related tasks, such as 'Faucet Close' and 'Faucet Open', as well as 'Window Open', 'Window Close', and 'Drawer Close'. This similarity reflects the inherent relationships between tasks, which may facilitate knowledge transfer during the learning process.

Furthermore, we also apply t-SNE to visualize the actions and joint embeddings of the 150 trajectories for each task. Each point in the visualization is color-coded with varying transparency to denote its temporal position within the trajectory, with points closer to the end being less transparent. The visualizations are shown in Figures 7 and 6. From this analysis, we find that the overall distributions of actions and joint embeddings across different tasks are similar, which partially explains the generalization capability of JEPT. Additionally, there are specific concentrated regions in the distribution for each task, which likely arise from the unique nature of each task.

Table 9: Success Rates (%) of JEPT injected with different visual priors calculated from 50 trials for each task in RoboSuite tasks. The average success rate of the tasks in $\mathcal{T}_{\text{demo}}$ and $\mathcal{T}_{\text{vid}}$ are listed in the first two rows, while the individual success rate of the 4 tasks in $\mathcal{T}_{\text{unseen}}$ are listed in the following rows. The average success rate of the seen and unseen tasks are calculated in the last two row.

| Task | JEPT+FlowFormer | JEPT+VideoMAE-v2 | JEPT+Dino-v2 | JEPT+SAM | JEPT |
|---|---|---|---|---|---|
| $\mathcal{T}_{\text{demo}}$ | **27.7** | 20.7 | 24.3 | 26.7 | 27.3 |
| $\mathcal{T}_{\text{vid}}$ | **13.2** | 11.6 | 10.8 | 8.4 | 12.8 |
| **Panda Lift** | 34.0 | 14.0 | 16.0 | 12.0 | **38.0** |
| **Sawyer Lift** | **18.0** | 0.0 | 4.0 | 2.0 | 16.0 |
| **IIWA Lift** | 8.0 | 8.0 | 0.0 | 0.0 | **12.0** |
| **UR5e Lift** | **14.0** | 0.0 | 0.0 | 0.0 | 8.0 |
| **Seen Average** | **20.4** | 16.1 | 17.6 | 17.5 | 20.1 |
| **Unseen Average** | **18.5** | 5.5 | 5.0 | 3.5 | **18.5** |

Table 10: Success Rates (%) of JEPT and the ablations on $c$ calculated from 50 trials for each task in Meta-World. The average success rates of the tasks in $\mathcal{T}_{\text{demo}}$ and $\mathcal{T}_{\text{vid}}$ are listed in the first two rows, while the individual success rates of the 4 tasks in $\mathcal{T}_{\text{unseen}}$ are listed in the following rows. The average success rates of the seen and unseen tasks are listed in the last two row.

| Task | $c = 0.3$ | $c = 1$ | $c = 5$ | $c = 8$(JEPT) | $c = 10$ | $c = 12$ |
|---|---|---|---|---|---|---|
| $\mathcal{T}_{\text{demo}}$ | 38.5 | 33.0 | 47.0 | 51.3 | 52.5 | **54.5** |
| $\mathcal{T}_{\text{vid}}$ | 19.0 | 17.7 | 24.3 | **31.7** | 27.7 | 12.3 |
| **Handle Press** | 4.0 | 12.0 | 24.0 | **28.0** | 14.0 | 6.0 |
| **Lever Pull** | 0.0 | 0.0 | 8.0 | 10.0 | **12.0** | 0.0 |
| **Plate Slide Back** | 2.0 | 8.0 | 6.0 | 14.0 | **16.0** | 2.0 |
| **Faucet Open** | 0.0 | 2.0 | 16.0 | **22.0** | 20.0 | 8.0 |
| **Seen Average** | 28.8 | 25.3 | 35.7 | **41.5** | 40.1 | 33.4 |
| **Unseen Average** | 1.5 | 5.5 | 13.5 | **18.5** | 15.5 | 4.0 |

Table 11: Average Success Rates (%) of JEPT variants on the 4 tasks in $\mathcal{T}_{\text{unseen}}$ in Meta-World experiments. The columns represent the variants with different ratios of $\mathcal{T}_{\text{demo}}$ and $\mathcal{T}_{\text{vid}}$, while the rows represent the variants with different values of hyperparameter $c$.

| $\mathcal{T}_{\text{demo}} + \mathcal{T}_{\text{vid}}$ | $(1) + (19)$ | $(3) + (17)$ | $(5) + (15)$ | $(8) + (12)$ | $(10) + (10)$ | $(12) + (8)$ |
|---|---|---|---|---|---|---|
| $c = 1$ | 0.0 | 0.0 | 8.0 | 5.0 | 3.0 | 9.0 |
| $c = 5$ | 0.0 | 1.0 | 5.5 | 7.5 | 10.0 | 8.5 |
| $c = 8$ | **1.5** | 2.0 | 3.5 | 8.5 | **18.0** | **18.5** |
| $c = 10$ | 0.0 | 0.0 | 6.0 | 10.0 | 16.0 | 10.0 |
| $c = 12$ | 0.5 | 2.5 | 8.0 | **19.5** | 14.5 | 9.5 |
| $c = 16$ | 0.0 | **5.0** | **12.5** | 12.0 | 10.0 | 10.5 |
| $c = 20$ | 1.0 | 0.0 | 10.0 | 11.0 | 8.0 | 4.0 |

Table 12: Success Rates (%) of JEPT and two baselines calculated from 50 trials for each task in Meta-World. The average success rate of the tasks in $\mathcal{T}_{\text{demo}}$ and $\mathcal{T}_{\text{vid}}$ are listed in the first two rows, while the individual success rate of the 4 tasks in $\mathcal{T}_{\text{unseen}}$ are listed in the following rows. The average success rates of the seen and unseen tasks are listed in the last two row.

| Task | BC+IDM | BC+IDM$^{\star}$ | JEPT |
|---|---|---|---|
| $\mathcal{T}_{\text{demo}}$ | 49.5 | 47.5 | **51.3** |
| $\mathcal{T}_{\text{vid}}$ | 8.3 | 14.7 | **31.7** |
| **Handle Press** | 8.0 | 7.0 | **28.0** |
| **Lever Pull** | 0.0 | 0.0 | **10.0** |
| **Plate Slide Back** | 0.0 | 0.0 | **14.0** |
| **Faucet Open** | 4.0 | 3.0 | **22.0** |
| **Seen Average** | 28.9 | 31.1 | **41.5** |
| **Unseen Average** | 3.0 | 2.5 | **18.5** |

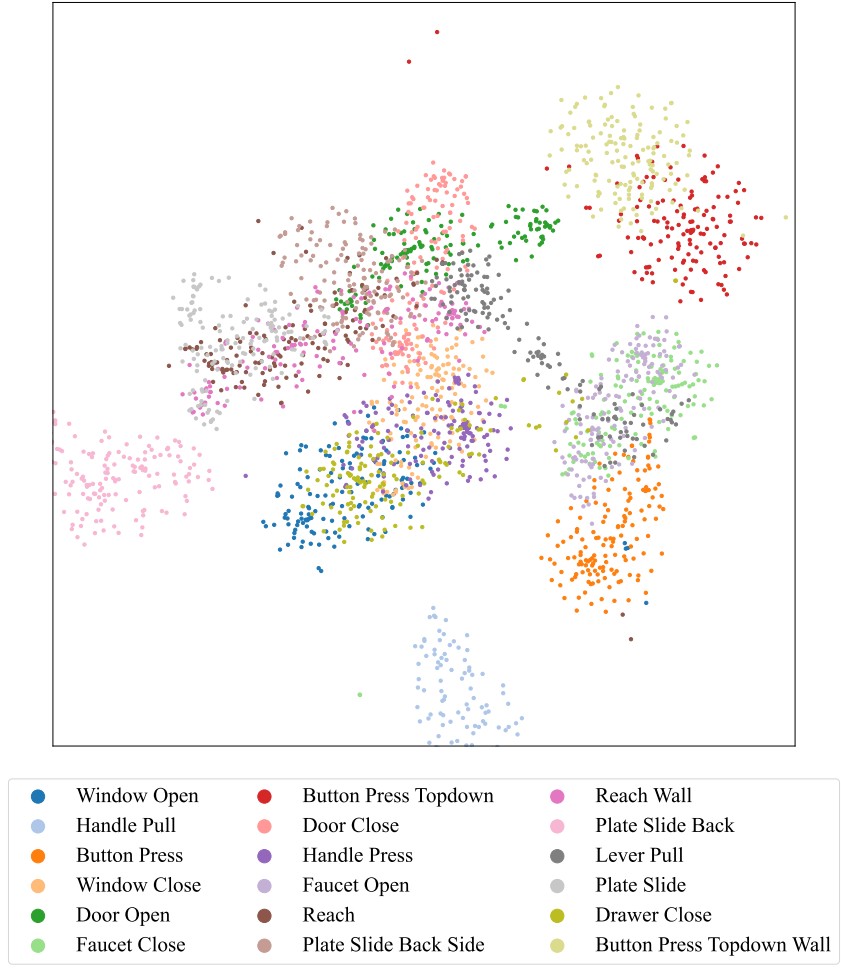

Figure 5: Visualization of the prompt video embeddings learned in JEPT. The prompt video embeddings are calculated from 150 videos of each task in Meta-World. We apply t-SNE on the whole embedding set.

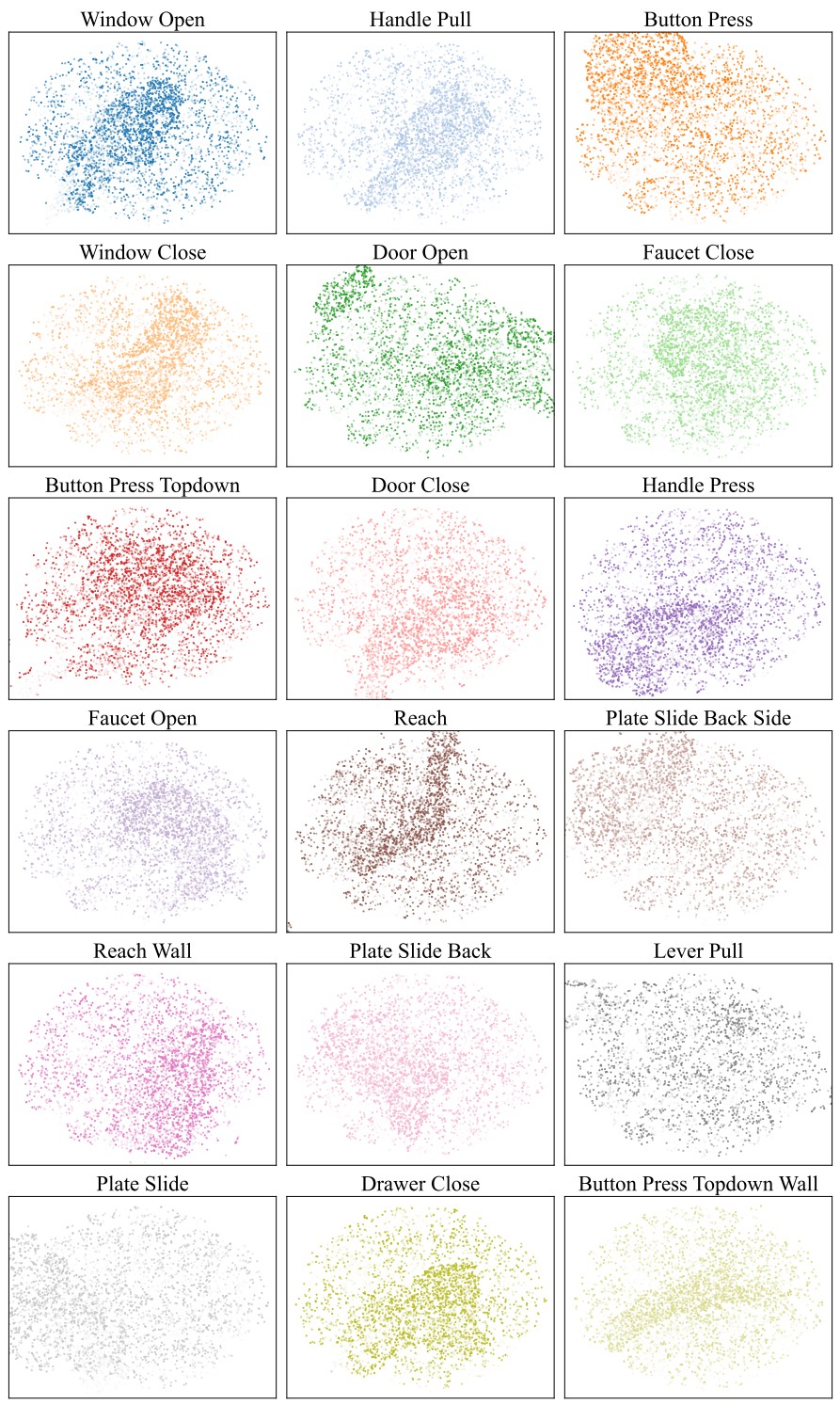

Figure 6: Visualization of the joint embeddings learned in JEPT. The joint embeddings are calculated from 150 trajectories of each task in Meta-World. We apply t-SNE on one whole embedding set and split the projected vectors according to the tasks into 18 sub-figures. The transparency of the points reflects the temporal positions of the corresponding embeddings, with points closer to the end being less transparent.

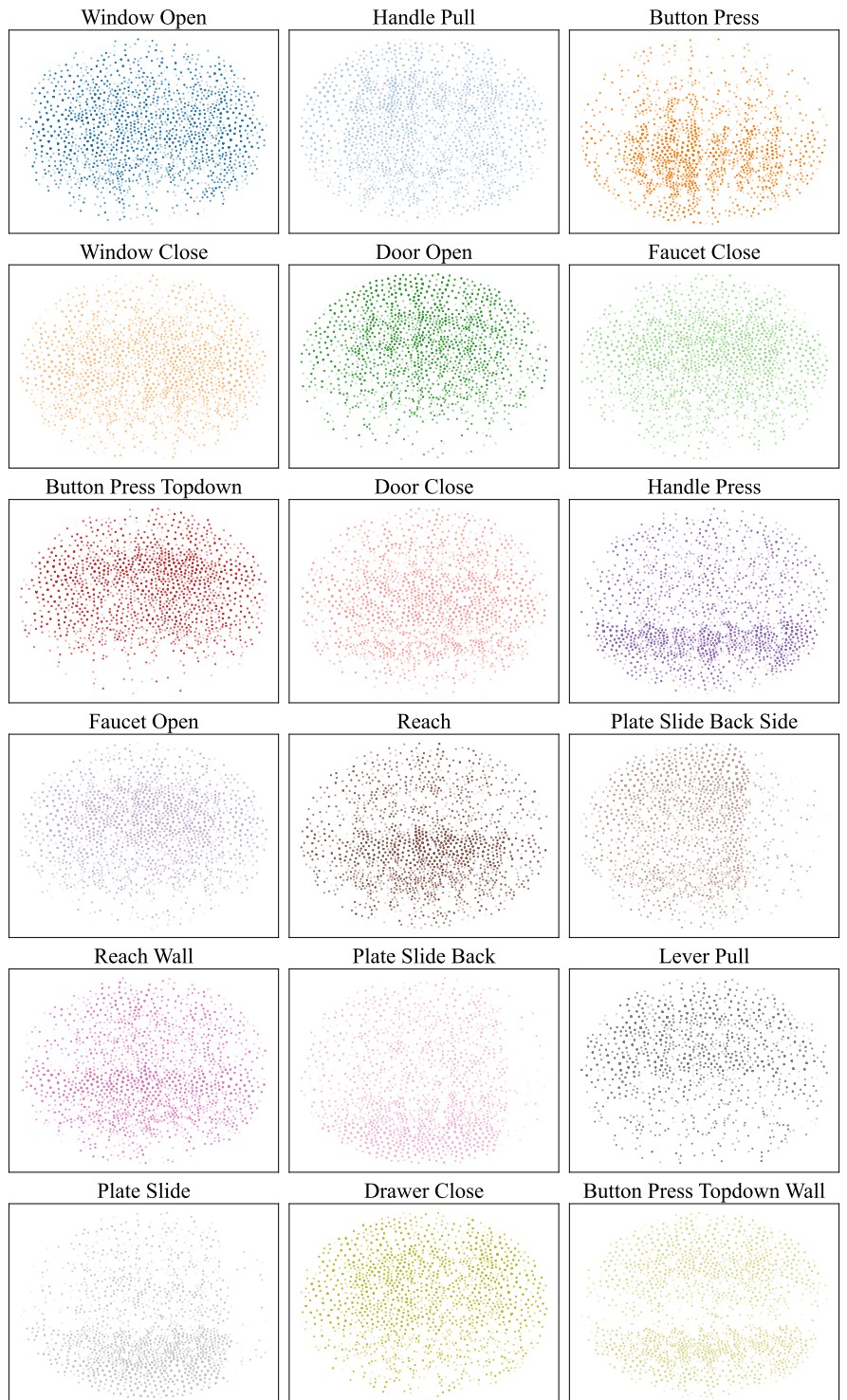

Figure 7: Visualization of the actions learned in JEPT. The actions are collected from 150 trajectories of each task in Meta-World. We apply t-SNE on one whole embedding set and split the projected vectors according to the tasks into 18 sub-figures. The transparency of the points reflects the temporal positions of the corresponding embeddings, with points closer to the end being less transparent.

