# OpenReview forum: "Learning Video-Conditioned Policy on Unlabelled Data with Joint Embedding Predictive Transformer"
_ICLR.cc/2025/Conference — ICLR 2025 Poster_

### Official Review · Reviewer_Zv2N · 2024-10-28

**Soundness:** 3
**Presentation:** 3
**Contribution:** 3
**Rating:** 6
**Confidence:** 2

**Summary:**

This paper introduces the Joint Embedding Predictive Transformer (JEPT), a novel approach for video-conditioned policy learning that leverages both expert demonstrations and expert videos paired with prompt videos to reduce the need for action label annotation. To effectively learn from the mixture datasets, JEPT decomposes the learning task into two subtasks: visual transition prediction and inverse dynamics learning. By jointly training these subtasks, JEPT enhances the generalization of the video-conditioned policy. Experimental results demonstrate that JEPT outperforms baseline methods.

**Strengths:**

1. The proposed JEPT framework for learning from mixed datasets is clear and reasonable.
2. Experiments on the Meta-World and Robosuite benchmarks demonstrate the effectiveness of JEPT.
3. Results indicate that visual priors can be integrated into JEPT to improve generalization.

**Weaknesses:**

1. Does BC+IDM train both the video encoder and the inverse dynamics model on demonstration videos? What if a fixed pretrained video encoder is used instead? This could ensure that the representations of demonstration and expert videos are in the same space, potentially aiding in the learning of generalizable actions.
2. How does JEPA influence performance? Did the authors experiment with other pretrained models, including different architectures and pretrained data?
3. The comparison of different visual prior injection methods appears inconsistent. The four models rely on distinct visual signals as inputs, which has led the authors to conclude that optical flow is the most suitable prior. However, these models were trained on different datasets and under varying conditions (self-supervised or supervised). It is crucial to detail these differences, as they may influence the performance.

**Questions:**

1. Does BC+IDM train both the video encoder and the inverse dynamics model on demonstration videos? What if a fixed pretrained video encoder is used?
2. How does JEPA influence performance? Did the authors experiment with other pretrained models, including different architectures and pretrained data?

---

> ### Author Response · Authors · 2024-11-23
> **Response to Reviewer Zv2N**
>
> We are thankful to the reviewer for their meticulous and in-depth review and for providing valuable insights. We are pleased to receive recognition for the effectiveness and the reasonability of our method. Next, we will respond to the weaknesses.
>
> >W1. Does BC+IDM train both the video encoder and the inverse dynamics model on demonstration videos? What if a fixed pretrained video encoder is used instead? This could ensure that the representations of demonstration and expert videos are in the same space, potentially aiding in the learning of generalizable actions.
>
> In the BC+IDM method, there are two video encoders involved: one is used for processing trajectories, and the other is used to encode the prompt video. From the context, the reviewer is likely referring to the former. Indeed, when learning IDM, the video encoder that processes trajectories is trained on demonstration data. It is reasonable to replace it with a fixed pre-trained video encoder to reduce distribution shift and thus potentially help generalize actions. Most pre-trained video encoders use a kernel size of 2 for temporal compression, which results in the representation length being halved in the temporal dimension. Similar to our approach in visual prior injection with VideoMAE v2, we repeat all frames within the video except for the first frame to ensure consistent length. Using VideoMAE v2, experimental results are shown in Appendix C.2. After switching to a fixed video encoder, BC+IDM is denoted as BC+IDM$^\star$. There is a slight improvement in success rates on unlabeled tasks, but there is no significant gain on unseen tasks. In all, the improvement from using a fixed video encoder is limited.
>
> | Task                        | BC+IDM | BC+IDM$^\star$ | JEPT     |
> | --------------------------- | ------ | -------------- | -------- |
> | $\mathcal{T}_{\text{demo}}$ | 49.5   | 47.5           | **51.3** |
> | $\mathcal{T}_{\text{vid}}$  | 8.3    | 14.7           | **31.7** |
> | Handle Press                | 8.0    | 7.0            | **28.0** |
> | Lever Pull                  | 0.0    | 0.0            | **10.0** |
> | Plate Slide Back            | 0.0    | 0.0            | **14.0** |
> | Faucet Open                 | 4.0    | 3.0            | **22.0** |
> | **Seen Average**            | 28.9   | 31.1           | **41.5** |
> | **Unseen Average**          | 3.0    | 2.5            | **18.5** |
>
>
> >W2. How does JEPA influence performance? Did the authors experiment with other pretrained models, including different architectures and pretrained data?
>
> There might be a slight misunderstanding at this point. We do not use a pre-trained JEPA; instead, we extend JEPA to a Transformer architecture for decision-making tasks and train it on the data we collect.
>
> >W3. The comparison of different visual prior injection methods appears inconsistent. The four models rely on distinct visual signals as inputs, which has led the authors to conclude that optical flow is the most suitable prior. However, these models were trained on different datasets and under varying conditions (self-supervised or supervised). It is crucial to detail these differences, as they may influence the performance.
>
> Thank the reviewer for pointing this out. In our visual prior experiments, the inputs for the four models are indeed inconsistent (single frame, two frames, multiple frames). Although all models use a visual Transformer architecture, there are differences in their structures (such as patch size, number of layers, and number of heads). This certainly warrants further explanation, and we have added this information in Appendix A.2.
>
> Additionally, as we mentioned in the paper, "The poor performance of other visual priors may result from the inherent unsuitability of these priors for visual control tasks or the inadequacy of the injection approach, requiring further exploration in future work." We don't conclude that optical flow is the most suitable prior for video-conditioned policy learning. It just performs better in our naive injection design. These visual encoders have been trained on large datasets across various visual tasks and possess certain generalizable priors for handling visual inputs. Our visual prior experiments aim to determine whether a joint embedding that predicts these representations with visual priors can further compress visual information and enhance generalization. From the perspective of introducing more generalizable visual priors, using different priors to process visual inputs makes sense. However, successfully learning a predictive joint embedding of these priors and effectively improving performance may depend on selecting more refined injection methods and better training method designs. We only reveal the potential of JEPT in this aspect, and further detailed research on other visual prior injections is left for future work.

---

> > ### Comment · Reviewer_Zv2N · 2024-11-24
> >
> > Thank you for providing additional experimental findings and clarifying the misunderstood points. The response addresses my concerns.

---

> > > ### Author Response · Authors · 2024-11-25
> > > **Response to Reviewer Zv2N**
> > >
> > > Dear Reviewer Zv2N,
> > >
> > > Thank you for your quick response. Your constructive feedback has been instrumental in enhancing our work.
> > >
> > > As the rebuttal deadline is approaching, we want to check if there are any other questions or issues you would like us to address. If there are no further concerns and considering that we have addressed your concerns, we would be grateful if you could consider providing us with stronger comments.
> > >
> > > Thank you once again for your support and guidance.
> > >
> > > Best regards,
> > >
> > > Authors of Submission6280

---

### Official Review · Reviewer_grHE · 2024-11-03

**Soundness:** 3
**Presentation:** 3
**Contribution:** 3
**Rating:** 6
**Confidence:** 3

**Summary:**

the paper propose Joint Embedding Predictive Transformer (JEPT), where the model can be conditioned on some video based demonstations, and the causal transformer also learns to predict the next action and observation (in pixels) space. The prompt video encoder, can be used to give demonstactions just from a video, this is great because giving an intertion is hard with language, and even babies do this from visual demonstations. The JEPA style video model, encodes causally the state and actions, and the joint encoder maps actions to states and there are jointly optimized during training. The paper evaulates these models on various benchmarks, and shows good performance.

**Strengths:**

The main stength of this paper is to able to prompt the robot from simple visual demonstrations. This has very nice benifits from not having any dependcy from langauges and from development physcology it make sense, as babies learn/ and repeat actions by watching others.

The paper has through experiments on meta world and robosuit, and in most of the cases the performance is good compared to previous works.

Ablations also clearly shows the benifits of joint embedding model.

**Weaknesses:**

more qualititative samples might be also helpful to see the experiments in action, and to show how the rollout of actions and h_t changes over time.

Lack of any real world examples is one weakness, it would be nice if the paper can show these nice propeties of giving a simple video demo, and the robot can mimic it in real world setting, that would be great, and can also answer many questions regarding the usefullness of the work.

It would be also nice, if the paper can address how the patch tokens are encoded, and how they are handled during the autoregresive stage of the pollicy.

**Questions:**

please look at my strengths and weakness sections, and if you can adress the weakness section, i am happy to change my ratings.

---

> ### Author Response · Authors · 2024-11-23
> **Response to Reviewer grHE**
>
> We appreciate the reviewer for the detailed and comprehensive reviews, and for offering insightful feedback. We are delighted to receive recognition for our method and experiments. Below, we will respond to the concerns raised by the reviewers.
>
> > W1. More qualititative samples might be also helpful to see the experiments in action, and to show how the rollout of actions and $h_t$ changes over time.
>
> Thanks for pointing this out. We add additional visualizations of the actions and $h_t$ in the rollouts in Appendix C.3 (Fig 7 and 6).
>
> >W2. Lack of any real world examples is one weakness, it would be nice if the paper can show these nice properties of giving a simple video demo, and the robot can mimic it in real world setting, that would be great, and can also answer many questions regarding the usefulness of the work.
>
> We sincerely thank the reviewer for highlighting this important point. We wholeheartedly agree that training on real-world datasets is vital and represents a significant challenge. Unfortunately, due to our limited experimental resources and the tight time constraints during the rebuttal period, we are unable to conduct this experiment at present. In our setting, the model is trained using aligned pairs of video prompts and expert video or demonstration data. For real-world environments, we do not yet have access to the necessary equipment to perform these experiments. Additionally, for simulated environments using real-world video prompts, we lack the aligned data needed between real-world video prompts and simulator environments. In fact, the experiments in Vid2Robot have demonstrated that simple designs can achieve visual imitation in the real world through training with a large amount of aligned data. We acknowledge the importance of real-world experiments and will address this in future research when we overcome these limitations. Thanks for the valuable feedback and understanding.
>
> >W3. It would be also nice, if the paper can address how the patch tokens are encoded, and how they are handled during the autoregressive stage of the policy.
>
> Thank the reviewer for pointing this out. In the main text, we use the term 'spatial encoder' as a shorthand to refer to this component. We have provided an explanation in the appendix. Specifically, we use a Vision Transformer (ViT) to encode the visual input and flatten all the patch tokens from the ViT to feed into the Perceiver IO, which compresses the number of tokens using learnable queries. The structure of the Perceiver IO is the same as in the original paper, with some hyperparameters modified. During the autoregressive phase, the compressed visual tokens are used as observation tokens for the current timestep and are input into the Video-Conditioned Causal Predictor. In the causal predictor, the causal mask is applied only temporally, allowing patch tokens within the same timestep to attend to each other. We have highlighted these parts in green in the Appendix.

---

> > ### Comment · Reviewer_grHE · 2024-11-24
> >
> > Thank you for providing the clarifications and providing samples. I also understand the time and resource constraints. The response addresses my concerns.

---

> > > ### Author Response · Authors · 2024-11-25
> > > **Response to Reviewer grHE**
> > >
> > > Dear Reviewer grHE,
> > >
> > > Thank you for your prompt response. Your constructive review has significantly helped us improve our work.
> > >
> > > As the rebuttal period is nearing its end, we want to check if there are any additional questions or issues you would like us to address. If there are no further concerns and given that we have addressed your concerns, we would appreciate it if you could consider giving us some stronger comments.
> > >
> > > Thank you once again for your support and guidance.
> > >
> > > Best regards,
> > >
> > > Authors of Submission6280

---

### Official Review · Reviewer_yXfU · 2024-11-07

**Soundness:** 3
**Presentation:** 2
**Contribution:** 3
**Rating:** 8
**Confidence:** 3

**Summary:**

A training strategy for video-conditioned policy generation is proposed in this manuscript. Considering the challenge of annotating action labels, the author propose to split the behavior cloning task into two sub tasks to reduce the dependencies on annoatated data. The visual transition prediction task predicts the observation embedding of next time step in the feature space, and the inverse dynamics leanring task estimates the current action by taking the consecutive two observations of current time step. The training of the visual transition prediction needs only the prompt videos and expert videos without action annotation. The training of inverse dynamics learning requires the prompt video, the expert video and the corresponding annotations. This training strategy is quite similar to the popular strategy pre-training (visual transition prediction) + supervised learning (inverse dynamics learning) . The proposed method is evaluated on two benchmark datasets, the Meta-World task and the Robosuite task, and the experimental results show the effectiveness of the proposed mtehod.

**Strengths:**

1. The proposed method aims to introduce prompt video as guidance for task imitation learning, which is interesting.
2. The involvement of visual transition prediction for observation embedding prediction is good. It can not only help reduce the cost of action annotation, but also make the task easier to learn. The prediction of observation embedding in the next time step in feature space is easier than the direct embedding extraction from the raw observation of the next time step. However, the premise the embedding space is well learned.
3. The proposed method is extensively evaluated and achieves good performance.

**Weaknesses:**

1. The paper is not easy to follow, especially for the researcher not in the robotics field. A further polishment of the manuscript would be great

* The authors are suggested to clarify each term such as prompt video (which is used as a whole global representation in the method), expert video. A simple explanation would be apprecaited;
* Based on my understanding, the key design of the proposed method is the split of behavior cloning into two sub-tasks the visual transition prediction and the inverse dynamics leanring, which can help utilize the mixture dataset, and reduce the dependency on action annoataion. However, the authors did not discuss much on why this design works and why it is suitable for the situation when the annotation of actions is limitted;

2. From the view of pretraining, a well pre-trained network needs tremendous unlabelled data to fit a good network. For video/image understanding, the requiremenet of large-scale data is easy to fullfill. However for the robotics task, the large-scale data is relatively diffcult to acquire.

3. Considering the diversity of robot configuration, the action state or the description of action in each robot is differernt. This paper does not mention how to handle the heterogeneity issue among robots, which limits the scenario span of the proposed method.

**Questions:**

1. How to handle the heterogeneity issue? Is the action space designed for only one robot configuration?
2. What is the ratio of $D_{demo}$ and $D_{vid}$ in the mixture dataset? From the view of model pretraining, does the $D_{vid}$ require much more samples than $D_{demo}$? How the ratio influence the final performance? An involvement of the ablation study on the ratio would be appreciated.
3. Is the hyperparameter $c$ in Eq.9 also influenced by the ratio of $D_{demo}$ and $D_{vid}$?
4. Considering $E_{prt}$ is a global representation of the prompt video, it would be interesting to see how $E_{prt}$ of videos from different action categories distribute in the feature space. Ideally, there should be high differentiation among the $E_{prt}$ from different action categories such that the model will not be misled by the guidance from $E_{prt}$.

---

> ### Author Response · Authors · 2024-11-23
> **Response to Reviewer yXfU (1/2)**
>
> We express our gratitude to the reviewer for the meticulous and thorough review, and for providing valuable insights. We are so pleased to receive recognition for the method and the experiments. Below, we will respond to the weaknesses and issues raised by the reviewers:
>
> >W1:  The paper is not easy to follow, especially for the researcher not in the robotics field. A further polishment of the manuscript would be great
> >- The authors are suggested to clarify each term such as prompt video (which is used as a whole global representation in the method), expert video. A simple explanation would be appreciated;
> >- Based on my understanding, the key design of the proposed method is the split of behavior cloning into two sub-tasks the visual transition prediction and the inverse dynamics learning, which can help utilize the mixture dataset, and reduce the dependency on action annotation. However, the authors did not discuss much on why this design works and why it is suitable for the situation when the annotation of actions is limited;
>
> Thanks for pointing this out. To more clearly present our setting, we have replaced the first figure with a clearer one and made some adjustments in the text. The core design of the algorithm works for two reasons: on one hand, visual transition prediction leverages the visual transition information in action-labeled and unlabeled expert videos; on the other hand, the generalization of inverse dynamics knowledge allows for inferring reasonable actions on unlabeled tasks. For a visual imitation learning task, visual transition prediction is the more task-specific part, requiring more data to learn, whereas inverse dynamics is more task-unified and can generalize even with limited action annotation. We have also made slight adjustments to emphasize this.
>
> > W2: From the view of pretraining, a well pre-trained network needs tremendous unlabelled data to fit a good network. For video/image understanding, the requiremenet of large-scale data is easy to fullfill. However for the robotics task, the large-scale data is relatively diffcult to acquire.
> >
> > W3: Considering the diversity of robot configuration, the action state or the description of action in each robot is different. This paper does not mention how to handle the heterogeneity issue among robots, which limits the scenario span of the proposed method.
> >
> > Q1. How to handle the heterogeneity issue? Is the action space designed for only one robot configuration?
>
> These three pieces of feedback seem to be related. From a data perspective, large-scale data for decision-making problems is emerging, such as Open-X and Vid2Robot. Our work, in fact, reduces the burden of constructing these large-scale datasets by using unlabeled data to expand the dataset for video-conditioned policy. When training on large-scale data, the heterogeneity issue absolutely needs to be taken into consideration. We appreciate the reviewer for pointing this out, which is very enlightening. In our setting, only one action space is included. For the mixed data from multiple action spaces, the impact on visual transition prediction is smaller because it focuses more on aligning with the prompt video at the visual level. The inverse dynamics part, which directly involves multiple action spaces, can be handled more crudely by merging them into a large action space and continuing to use a shared network for learning, or more finely by equipping each action space with a decoder head to achieve mapping to different spaces. Of course, this is only a possible extension of our framework for data with multiple action spaces, and the specific effectiveness needs to be experimentally verified in future work. Due to our limited experimental resources and the time constraints during the rebuttal period, we are currently unable to conduct experiments in this aspect. We do thank the reviewer for the very enlightening suggestion. Our current work focuses on incorporating unlabeled data into video-conditioned policy, and we will further consider expanding our training dataset from this perspective.

---

> ### Author Response · Authors · 2024-11-23
> **Response to Reviewer yXfU (2/2)**
>
> > Q2. What is the ratio of $D_{demo}$ and $D_{vid}$ in the mixture dataset? From the view of model pretraining, does the $D_{vid}$ require much more samples than $D_{demo}$? How does the ratio influence the final performance? An involvement of the ablation study on the ratio would be appreciated.
>
> > Q3. Is the hyperparameter $c$ in Eq.9 also influenced by the ratio of $D_{demo}$ and $D_{vid}$?
>
> This is an excellent aspect of the ablation study, and we thank the reviewer for pointing it out. In the original ablation, we adjusted the number of tasks in $D_{vid}$​ to change its size, which alters the amount of data used for training to some extent. To address this, we conduct supplementary experiments, adding data from 6 more tasks in Metaworld, adjusting the ratio of $T_{vid}$​ and $T_{demo}$​ across a total of 20 training tasks, and testing different values of hyperparameter $c$ in Eq.9 under different ratios. We report the average success rate on unseen tasks, and the results are updated in Appendix C.2.
>
> | $T_{\text{demo}}+T_{\text{vid}}$ | $(1) + (19)$ | $(3) + (17)$ | $(5) + (15)$ | $(8) + (12)$ | $(10) + (10)$ | $(12) + (8)$ |
> | ---------------------------------------------------- | ------------ | ------------ | ------------ | ------------ | ------------- | ------------ |
> | $c = 1$                                              | 0.0          | 0.0          | 8.0          | 5.0          | 3.0           | 9.0          |
> | $c = 5$                                              | 0.0          | 1.0          | 5.5          | 7.5          | 10.0          | 8.5          |
> | $c = 8$                                              | **1.5**      | 2.0          | 3.5          | 8.5          | **18.0**      | **18.5**     |
> | $c = 10$                                             | 0.0          | 0.0          | 6.0          | 10.0         | 16.0          | 10.0         |
> | $c = 12$                                             | 0.5          | 2.5          | 8.0          | **19.5**     | 14.5          | 9.5          |
> | $c = 16$                                             | 0.0          | **5.0**      | **12.5**     | 12.0         | 10.0          | 10.5         |
> | $c = 20$                                             | 1.0          | 0.0          | 10.0         | 11.0         | 8.0           | 4.0          |
>
> When the number of  $T_{demo}$​ is within a certain range (5-12), the ratio has little impact on success rate. When the number of  $T_{demo}$​ is too small, it is difficult to learn inverse knowledge sufficiently from the extremely limited action annotations, thus affecting the success rate. Additionally, when the proportion of $D_{vid}$​ increases, a larger $c$ may be needed to achieve good performance.
>
> > Q4. Considering $E_{prt}$ is a global representation of the prompt video, it would be interesting to see how $E_{prt}$ of videos from different action categories distribute in the feature space. Ideally, there should be high differentiation among the $E_{prt}$ from different action categories such that the model will not be misled by the guidance from $E_{prt}$.
>
> We encoded the video prompts from 18 Metaworld tasks using the video encoder learned by JEPT, flattened these $E_{prt}$​, and visualized them using t-SNE. The figure is put in Appendix C.3 (Figure 5). Indeed, clear differences among different tasks can be observed, indicating that the model can learn distinctions between different tasks to avoid misleading.  Additionally, we observe a degree of similarity between related tasks, such as ‘Faucet Close’ and ‘Faucet Open’, as well as ‘Window Open’, ‘Window Close’, and ‘Drawer Close’. This similarity reflects the inherent relationships between tasks, which may facilitate knowledge transfer during the learning process.

---

> > ### Comment · Reviewer_yXfU · 2024-11-25
> >
> > Thanks for the rebuttal. I have no further concern.

---

> > > ### Author Response · Authors · 2024-11-25
> > > **Response to Reviewer yXfU**
> > >
> > > Dear Reviewer yXfU,
> > >
> > > Thank you for your quick response. Your insightful reviews have been incredibly helpful in enhancing our work!
> > >
> > > Thank you once again for your support and guidance.
> > >
> > > Best regards,
> > >
> > > Authors of Submission6280

---

### Official Review · Reviewer_4tQo · 2024-11-08

**Soundness:** 3
**Presentation:** 3
**Contribution:** 3
**Rating:** 8
**Confidence:** 3

**Summary:**

Video-conditioned policy learning often requires extensive action-labeled demonstrations, which are costly and time-consuming to acquire.  To address this problem, a method named Joint Embedding Predictive Transformer (JEPT) is introduced in this paper. JEPT jointly
learns visual transition prediction, which predicts the next visual state in a sequence based on the current observation and prompt video with unlabelled data, and inverse dynamics, which learns to infer actions that cause the transition between two states given the labeled data. The model is evaluated on two benchmarks, Meta-World and Robosuite, and is compared with recent state-of-the-art methods. Experimental results show that JEPT achieves superior performances.

**Strengths:**

- The paper is novel and addresses real-world problems. More specifically, decomposing the behavior cloning into visual transition prediction and inverse dynamics learning is novel, prior approaches mostly rely on labeled data or direct behavior cloning, while JEPT makes use of unlabelled data and largely improved generalization ability and performance.
- The experiments are solid and thorough. JEPT is compared with recent state-of-the-art methods. The robustness analysis and visual prior injection studies are interesting and inspiring.

**Weaknesses:**

- The model’s performance appears sensitive to the choice of visual priors. It is unclear how to select desired priors across different environments, or when facing a new task how to choose the prior.
- The paper primarily validates JEPT on simulated datasets. The paper can be improved by adding real-world datasets for more complex environments, strengthening the claims of generalizability and practical applicability.

**Questions:**

Please refer to the Weaknesses section.

---

> ### Author Response · Authors · 2024-11-23
> **Response to Reviewer 4tQo**
>
> We are grateful to the reviewer for the meticulous and thorough review, and for providing insightful feedback. We are pleased to receive recognition for the comments such as 'novel', 'address real-world problems', and 'solid and thorough experiments'. Below, we will respond to the weaknesses raised by the reviewers:
>
> > W1:  The model’s performance appears sensitive to the choice of visual priors. It is unclear how to select desired priors across different environments, or when facing a new task how to choose the prior.
>
> To better illustrate the impact of visual prior injection across different environments, we conduct additional experiments on robosuite, replicating the visual prior injection method used in the meta-world experiment. The results, updated in Appendix C.1 and listed below, suggest that using visual encoders trained for optical flow estimate tasks still yields better results. This somewhat indicates that using optical flow can be a choice for different environments in our injection implementation. As we have mentioned in the paper, 'The poor performance of other visual priors may result from the inherent unsuitability of these priors for visual control tasks or the inadequacy of the injection approach, requiring further exploration in future work.' These visual encoders have been trained on large amounts of data across many visual tasks, and possess certain generalizable priors in handling visual inputs. Our visual prior experiments aim to determine whether a joint embedding predictive of these representations with visual priors can further compress visual information and enhance generalization. From the perspective of introducing more generalizable visual priors, these different priors for processing visual inputs ought to make sense. However, learning predictive joint embedding of these priors and effectively improving performance may depend on the choice of more refined injection methods and better design of training methods. We only reveal the potential of JEPT in this aspect, and further detailed research on other visual prior injections is left for future work.
>
> | Task                            | JEPT+FlowFormer | JEPT+VideoMAE-v2 | JEPT+Dino-v2 | JEPT+SAM | JEPT     |
> | ------------------------------- | --------------- | ---------------- | ------------ | -------- | -------- |
> | $ \mathcal{T}_{\text{demo}} $ | **27.7**        | 20.7             | 24.3         | 26.7     | 27.3     |
> | $ \mathcal{T}_{\text{vid}} $  | **13.2**        | 11.6             | 10.8         | 8.4      | 12.8     |
> | Panda Lift                      | 34.0            | 14.0             | 16.0         | 12.0     | **38.0** |
> | Sawyer Lift                     | **18.0**        | 0.0              | 4.0          | 4.0      | 16.0     |
> | IIWA Lift                       | 8.0             | 8.0              | 0.0          | 0.0      | **12.0** |
> | UR5e Lift                       | **14.0**        | 0.0              | 0.0          | 0.0      | 8.0      |
> | **Seen Average**                | **20.4**        | 16.1             | 17.6         | 17.5     | 20.1     |
> | **Unseen Average**              | **18.5**        | 5.5              | 5.0          | 3.5      | **18.5** |
>
>
> >W2: The paper primarily validates JEPT on simulated datasets. The paper can be improved by adding real-world datasets for more complex environments, strengthening the claims of generalizability and practical applicability.
>
> We sincerely thank the reviewer for highlighting this important point. We fully acknowledge the critical role that training on real-world datasets plays in advancing our field. Unfortunately, due to our current limitations in experimental resources and the time constraints of the rebuttal period, we are unable to conduct such experiments at this time. Our current model is trained using aligned pairs of video prompts and expert video or demonstration data. For real-world environments, we lack the necessary equipment to run experiments. Similarly, for simulated environments using real-world video prompts, we lack the aligned data necessary to bridge real-world video prompts with simulator environments. Our research focuses on introducing a training method that integrates demonstrations with unlabeled expert videos. We recognize that utilizing extensive real-world video data could significantly enhance our work, and we are eager to explore this avenue in future studies.

---

> > ### Comment · Reviewer_4tQo · 2024-11-25
> >
> > Thank you for the explanations of visual prior. I don't have further questions.

---

> ### Author Response · Authors · 2024-11-25
> **Looking Forward to the Feedback on the Rebuttal Response**
>
> Dear Reviewer 4tQo,
>
> We sincerely appreciate your invaluable feedback, which plays a crucial role in enhancing the quality of our work.
>
> As the rebuttal deadline approaches, we want to check whether our response addresses your concerns. If there are any additional questions or issues you would like us to address, please let us know. We would be grateful if you could provide stronger comments to support our work after reviewing our response.
>
> Thank you once again for your support and guidance.
>
> Best regards,
>
> Authors of Submission6280

---

### Meta-Review · Area_Chair_DCoV · 2024-12-20

**Metareview:**

(a) Scientific Claims and Findings

The paper introduces the Joint Embedding Predictive Transformer (JEPT), an approach for video-conditioned policy learning that reduces the need for action-labeled demonstrations. JEPT decomposes the learning task into visual transition prediction and inverse dynamics learning, leveraging both labeled and unlabeled data to enhance generalization. The model is evaluated on Meta-World and Robosuite benchmarks, showing better performance compared to state-of-the-art methods. Reviewers highlight the method's ability to utilize unlabelled data effectively and its potential to improve generalization in video-conditioned policy learning.

(b) Strengths

Reviewer 4tQo acknowledges the novelty of decomposing behavior cloning into two sub-tasks, which improves generalization and performance. yXfU finds the introduction of prompt video guidance and visual transition prediction interesting and effective in reducing annotation costs. grHE appreciates the ability to prompt robots from visual demonstrations, aligning with developmental psychology insights. Zv2N commends the clear and reasonable framework of JEPT and its demonstrated effectiveness on benchmarks.

(c) Weaknesses

The reviewers identify several weaknesses. 4tQo notes the model's sensitivity to visual priors and the lack of real-world dataset validation. yXfU suggests further polishing for clarity and questions the handling of heterogeneity among robots. grHE points out the absence of real-world examples and the need for more qualitative samples. Zv2N raises concerns about the training of video encoders and the consistency of visual prior injection methods, suggesting that differences in training conditions may affect performance.

(d) Decision Reasons

The AC agrees with the reviewer's recommendation to accept the paper. The final decision to accept the paper is based on its novel approach to video-conditioned policy learning and the strong empirical results demonstrated on benchmark datasets. The method's ability to leverage unlabelled data and improve generalization is a contribution, as highlighted by reviewers 4tQo and yXfU. While there are concerns about the model's sensitivity to visual priors and the lack of real-world validation, the overall strengths in innovation, experimental validation, and potential impact on the field outweigh these weaknesses. The paper's contributions to reducing annotation costs and enhancing policy learning make it a valuable addition to the conference.

**Additional Comments On Reviewer Discussion:**

During the rebuttal period, the authors effectively addressed the concerns raised by the reviewers, leading to a positive consensus.

Reviewer 4tQo expressed gratitude for the explanations provided regarding visual priors and indicated that they had no further questions, suggesting satisfaction with the authors' clarifications.

Reviewer yXfU thanked the authors for the rebuttal and confirmed that they had no further concerns, indicating that the authors' responses were satisfactory.

Reviewer grHE appreciated the clarifications and the provision of additional samples. They acknowledged the time and resource constraints faced by the authors and stated that their concerns were addressed, showing a positive reception to the authors' efforts.

Reviewer Zv2N thanked the authors for the additional experimental findings and clarifications, noting that their concerns were addressed.
This indicates that the authors successfully resolved any misunderstandings and provided the necessary information to satisfy the reviewer.

In weighing these points for the final decision, the authors' ability to address all reviewer concerns effectively during the rebuttal period was a significant factor. The positive feedback from all reviewers, who acknowledged that their concerns were resolved, reinforced the decision to accept the paper.

---

### Decision · Program_Chairs · 2025-01-22

Accept (Poster)